# Learning Safe Multi-Agent Control with Decentralized Neural Barrier Certificates

**Zengyi Qin**[1]**, Kaiqing Zhang**[2]**, Yuxiao Chen**[3]**, Jingkai Chen**[1] **and Chuchu Fan**[1]
[1]Massachusetts Institute of Technology
[2]University of Illinois Urbana-Champaign
[3]California Institute of Technology
`{qinzy, chuchu}@mit.edu`

## Abstract

We study the multi-agent safe control problem where agents should avoid collisions to static obstacles and collisions with each other while reaching their goals. Our core idea is to learn the multi-agent control policy *jointly* with learning the control barrier functions as *safety certificates*. We propose a new joint-learning framework that can be implemented in a *decentralized* fashion, which can adapt to an arbitrarily large number of agents. Building upon this framework, we further improve the scalability by incorporating neural network architectures that are invariant to the quantity and permutation of neighboring agents. In addition, we propose a new spontaneous policy refinement method to further enforce the certificate condition during testing. We provide extensive experiments to demonstrate that our method significantly outperforms other leading multi-agent control approaches in terms of maintaining safety and completing original tasks. Our approach also shows substantial generalization capability in that the control policy can be trained with 8 agents in one scenario, while being used on other scenarios with up to 1024 agents in complex multi-agent environments and dynamics. Videos and source code can be found on the website[1].

## 1 Introduction

Machine learning (ML) has created unprecedented opportunities for achieving full autonomy. However, learning-based methods in autonomous systems (AS) can and do fail due to the lack of formal guarantees and limited generalization capability, which poses significant challenges for developing safety-critical AS, especially large-scale multi-agent AS, that are provably dependable.

On the other side, safety certificates (Chang et al. (2019); Jin et al. (2020); Choi et al. (2020)), which widely exist in control theory and formal methods, serve as proofs for the satisfaction of the desired properties of a system, under certain control policies. For example, once found, a Control Barrier Function (CBF) ensures that the closed-loop system always stays inside some safe set (Wieland & Allgöwer, 2007; Ames et al., 2014) with a CBF Quadratic Programming (QP) supervisory controller. However, it is extremely difficult to synthesize CBF by hand for complex dynamic systems, which stems a growing interest in learning-based CBF (Saveriano & Lee, 2020; Srinivasan et al., 2020; Jin et al., 2020; Boffi et al., 2020; Taylor et al., 2020; Robey et al., 2020). However, all of these studies only concern single-agent systems. How to develop learning-based approaches for safe multi-agent control that are both provably dependable and scalable remains open.

In multi-agent control, there is a constant dilemma: centralized control strategies can hardly scale to a large number of agents, while decentralized control without coordination often misses safety and performance guarantees. In this work, we propose a novel learning framework that *jointly* designs multi-agent control policies and safety certificate from data, which can be implemented in a *decentralized* fashion and scalable to an arbitrary number of agents. Specifically, we first introduce the notion of decentralized CBF as safety certificates, then propose the framework of learning decentralized CBF, with generalization error guarantees. The decentralized CBF can be seen as a

---

[1]https://realm.mit.edu/blog/learning-safe-multi-agent-control-decentralized-neural-barrier-certificates

contract among agents, which allows agents to learn a mutual agreement with each other on how to avoid collisions. Once such a controller is achieved through the joint-learning framework, it can be applied on an arbitrarily number of agents and in scenarios that are different from the training scenarios, which resolves the fundamental scalability issue in multi-agent control. We also propose several effective techniques in Section 4 to make such a learning process even more scalable and practical, which are then validated extensively in Section 5.

Experimental results are indeed promising. We study both 2D and 3D safe multi-agent control problems, each with several distinct environments and complex nonholonomic dynamics. Our joint-learning framework performs exceptionally well: our control policies trained on scenarios with 8 agents can be used on up to 1024 agents while maintaining low collision rates, which has notably pushed the boundary of learning-based safe multi-agent control. Speaking of which, 1024 is not the limit of our approach but rather due to the limited computational capability of our laptop used for the experiments. We also compare our approach with both leading learning-based methods (Lowe et al., 2017; Zhang & Bastani, 2019; Liu et al., 2020) and traditional planning methods (Ma et al., 2019; Fan et al., 2020). Our approach outperforms all the other approaches in terms of both completing the tasks and maintaining safety.

**Contributions.** Our main contributions are three-fold: 1) We propose the first framework to jointly learning safe multi-agent control policies and CBF certificates, in a decentralized fashion. 2) We present several techniques that make the learning framework more effective and scalable for practical multi-agent systems, including the use of quantity-permutation invariant neural network architectures in learning to handle the permutation of neighbouring agents. 3) We demonstrate via extensive experiments that our method significantly outperforms other leading methods, and has exceptional generalization capability to unseen scenarios and an arbitrary number of agents, even in quite complex multi-agent environments such as ground robots and drones. The video that demonstrates the outstanding performance of our method can be found in the supplementary material.

*Related Work.* **Learning-Based Safe Control via CBF.** Barrier certificates (Prajna et al., 2007) and CBF (Wieland & Allgöwer, 2007) is a well-known effective tool for guaranteeing the safety of nonlinear dynamic systems. However, the existing methods for constructing CBFs either rely on specific problem structures (Chen et al., 2017b) or do not scale well (Mitchell et al., 2005). Recently, there has been an increasing interest in learning-based and data-driven safe control via CBFs, which primarily consist of two categories: *learning CBFs from data* (Saveriano & Lee, 2020; Srinivasan et al., 2020; Jin et al., 2020; Boffi et al., 2020), and *CBF-based approach for controlling unknown systems* (Wang et al., 2017; 2018; Cheng et al., 2019; Taylor et al., 2020). Our work is more pertinent to the former and is complementary to the latter, which usually assumes that the CBF is provided. None of these learning-enabled approaches, however, has addressed the multi-agent setting.

**Multi-Agent Safety Certificates and Collision Avoidance.** Restricted to holonomic systems, guaranteeing safety in multi-agent systems has been approached by limiting the velocities of the agents (Van den Berg et al., 2008; Alonso-Mora et al., 2013). Later, Borrmann et al. (2015) Wang et al. (2017) have proposed the framework of *multi-agent CBF* to generate collision-free controllers, with either perfectly known system dynamics (Borrmann et al., 2015), or with worst-case uncertainty bounds (Wang et al., 2017). Recently, Chen et al. (2020) has proposed a *decentralized* controller synthesized approach under this CBF framework, which is scalable to an arbitrary number of agents. However, in Chen et al. (2020) the CBF controller relies on online integration of the dynamics under the backup strategy, which can be computationally challenging for complex systems. Due to space limit, we omit other non-learning multi-agent control methods but acknowledge their importance.

**Safe Multi-Agent (Reinforcement) Learning (MARL).** Safety concerns have drawn increasing attention in MARL, especially with the applications to safety-critical multi-agent systems (Zhang & Bastani, 2019; Qie et al., 2019; Shalev-Shwartz et al., 2016). Under the CBF framework, Cheng et al. (2020) considered the setting with *unknown* system dynamics, and proposed to design robust multi-agent CBFs based on the *learned* dynamics. This mirrors the second category mentioned above in single-agent learning-based safe control, which is perpendicular to our focus. RL approaches have also been applied for multi-agent collision avoidance (Chen et al., 2017a; Lowe et al., 2017; Everett et al., 2018; Zhang et al., 2018). Nonetheless, no formal guarantees of safety were established in these works. One exception is Zhang & Bastani (2019), which proposed a multi-agent model predictive shielding algorithm that provably guarantees safety for any policy learned from MARL, which differs from our multi-agent CBF-based approach. More importantly, none of these MARL-

based approaches scale to a massive number of, e.g., thousands of agents, as our approach does. The most scalable MARL platform, to the best of our knowledge, is Zheng et al. (2017), which may handle a comparable scale of agents as ours, but with *discrete* state-action spaces. This is in contrast to our continuous-space models that can model practical control systems such as robots and drones.

## 2 PRELIMINARIES

### 2.1 CONTROL BARRIER FUNCTIONS AS SAFETY CERTIFICATES

One common approach for (single-agent) safety certificate is via control barrier functions (Ames et al., 2014), which can enforce the states of dynamic systems to stay in the safe set. Specifically, let $\mathcal{S} \subset \mathbb{R}^n$ be the state space, $\mathcal{S}_d \subset \mathcal{S}$ is the dangerous set, $\mathcal{S}_s = \mathcal{S} \backslash \mathcal{S}_d$ is the safe set, which contains the set of initial conditions $S_0 \subset S_s$. Also define the space of control actions as $\mathcal{U} \subset \mathbb{R}^m$. For a dynamic system $\dot{s}(t) = f(s(t), u(t))$, a control barrier function $h : \mathbb{R}^n \mapsto \mathbb{R}$ satisfies:

$$(\forall s \in \mathcal{S}_0, h(s) \geq 0) \bigwedge (\forall s \in \mathcal{S}_d, h(s) < 0) \bigwedge (\forall s \in \{s \mid h(s) \geq 0\}, \nabla_s h \cdot f(s, u) + \alpha(h) \geq 0), \quad (1)$$

where $\alpha(\cdot)$ is a class-$\mathcal{K}$ function, i.e., $\alpha(\cdot)$ is strictly increasing and satisfies $\alpha(0) = 0$. For a control policy $\pi : \mathcal{S} \to \mathcal{U}$ and CBF $h$, it is proved in Ames et al. (2014) that if $s(0) \in \{s \mid h(s) \geq 0\}$ and the three conditions in (1) are satisfied with $u = \pi(x)$, then $s(t) \in \{s \mid h(s) \geq 0\}$ for $\forall t \in [0, \infty)$, which means the state would never enter the dangerous set $\mathcal{S}_d$ under $\pi$.

### 2.2 SAFETY OF MULTI-AGENT DYNAMIC SYSTEMS

Consider a multi-agent system with $N$ agents, the joint state of which at time $t$ is denoted by $s(t) = \{s_1(t), s_2(t), \cdots, s_N(t)\}$ where $s_i(t) \in \mathcal{S}_i \subset \mathbb{R}^n$ denotes the state of agent $i$ at time $t$. The dynamics of agent $i$ is $\dot{s}_i(t) = f_i(s_i(t), u_i(t))$ where $u_i(t) \in \mathcal{U}_i \subset \mathbb{R}^m$ is the control action of agent $i$. The overall state space and input space are denoted as $\mathcal{S} \doteq \bigotimes_{i=1}^N \mathcal{S}_i$, $\mathcal{U} \doteq \bigotimes_{i=1}^N \mathcal{U}_i$. For each agent $i$, we define $\mathcal{N}_i(t)$ as the set of its neighborhood agents at time $t$. Let $o_i(t) \in \mathbb{R}^{n \times |\mathcal{N}_i(t)|}$ be the local observation of agent $i$, which is the states of $|\mathcal{N}_i(t)|$ neighborhood agents. Notice that the dimension of $o_i(t)$ is not fixed and depends on the quantity of neighboring agents. We assume that the safety of agent $i$ is jointly determined by $s_i$ and $o_i$. Let $\mathcal{O}_i$ be the set of all possible observations and $\mathcal{X}_i := \mathcal{S}_i \times \mathcal{O}_i$ be the state-observation space that contains the safe set $\mathcal{X}_{i,s}$, dangerous set $\mathcal{X}_{i,d}$ and initial conditions $\mathcal{X}_{i,0} \subset \mathcal{X}_{i,s}$. Let $d : \mathcal{X}_i \to \mathbb{R}$ describe the minimum distance from agent $i$ to other agents that it observes, $d(s_i, o_i) < \kappa_s$ implies collision. Then $\mathcal{X}_{i,s} = \{(s_i, o_i) | d(s_i, o_i) \geq \kappa_s\}$ and $\mathcal{X}_{i,d} = \{(s_i, o_i) | d(s_i, o_i) < \kappa_s\}$. Let $\bar{d}_i : \mathcal{S} \to \mathbb{R}$ be the lifting of $d$ from $\mathcal{X}_i$ to $\mathcal{S}$, which is well-defined since there is a surjection from $\mathcal{S}$ to $\mathcal{X}_i$. Then define $\mathcal{S}_s \doteq \{s \in \mathcal{S} | \forall i = 1, ..., N, \bar{d}_i(s) \geq \kappa_s\}$. The safety of a multi-agent system can be formally defined as follows:

**Definition 1** (Safety of Multi-Agent Systems). *If the state-observation satisfies $d(s_i, o_i) \geq \kappa_s$ for agent $i$ and time $t$, then agent $i$ is safe at time $t$. If for $\forall i$, agent $i$ is safe at time $t$, then the multi-agent system is safe at time $t$, and $s \in \mathcal{S}_s$.*

A main objective of this paper is to learn the control policy $\pi_i(s_i(t), o_i(t))$ for $\forall i$ such that the multi-agent system is safe. The control policy is decentralized (i.e., each agent has its own control policy and there does not exist a central controller to coordinate all the agents). In this way, our decentralized approach has the hope to scale to very a large number of agents.

## 3 LEARNING FRAMEWORK FOR MULTI-AGENT DECENTRALIZED CBF

### 3.1 DECENTRALIZED CONTROL BARRIER FUNCTIONS

For a multi-agent dynamic system, the most naïve CBF would be a centralized function taking into account the cross production of all agents' states, which leads to an exponential blow-up in the state space and difficulties in modeling systems with an arbitrary number of agents. Instead, we consider a decentralized control barrier function $h_i : \mathcal{X}_i \mapsto \mathbb{R}$:

$$(\forall (s_i, o_i) \in \mathcal{X}_{i,0}, h_i(s_i, o_i) \geq 0) \bigwedge (\forall (s_i, o_i) \in \mathcal{X}_{i,d}, h_i(s_i, o_i) < 0) \bigwedge$$
$$(\forall (s_i, o_i) \in \{(s_i, o_i) \mid h_i(s_i, o_i) \geq 0\}, \nabla_{s_i} h_i \cdot f_i(s_i, u_i) + \nabla_{o_i} h_i \cdot \dot{o}_i(t) + \alpha(h_i) \geq 0) \quad (2)$$

where $\dot{o}_i(t)$ is the time derivative of the observation, which depends on the behavior of other agents. Although there is no explicit expression of this term, it can be evaluated and incorporated in the learning process. Note that the CBF $h_i(s_i, o_i)$ is local in the sense that it only depends on the local state $s_i$ and observation $o_i$. We refer to the three conditions in (2) as *decentralized CBF conditions*. The following proposition shows that satisfying (2) guarantees the safety of the multi-agent system.

**Proposition 1** (Multi-Agent Safety Certificates with Decentralized CBF). *If for $\forall i$, the initial state-observation $(s_i(0), o_i(0)) \in \{(s_i, o_i) \mid h_i(s_i, o_i) \geq 0\}$ and the decentralized CBF conditions in (2) are satisfied, then $\forall i$ and $\forall t$, $(s_i(t), o_i(t)) \in \{(s_i, o_i) \mid h_i(s_i, o_i) \geq 0\}$, which implies the state would never enter $\mathcal{X}_{i,d}$ for any agent $i$. Thus, by Definition 1, the multi-agent system is safe.*

The proof of Proposition 1 is provided in the supplementary material. The key insight of Proposition 1 is that for the whole multi-agent system, the CBFs can be applied in a *decentralized* fashion for each agent. Since $h_i(s_i, o_i) \geq 0$ is invariant, by definition of $h_i$, $h_i(s_i, o_i) > 0 \implies \bar{d}_i(s) \geq \kappa_s$, which means agent $i$ never gets closer than $\kappa_s$ to all its neighborhood agents. Therefore, $\forall i, h_i(s_i, o_i) \geq 0$ implies that $\forall i, \bar{d}_i(s) \geq \kappa_s$, which by definition also means $s \in \mathcal{S}_s$, and the multi-agent system is safe as defined in Definition 1.

Notice that an agent only needs to care about its local information, and if all agents respect the same form of contract (i.e., the decentralized CBF conditions), the whole multi-agent system will be safe. The fact that global safety can be guaranteed by decentralized CBF is of great importance since it reveals that a centralized controller that coordinates all agents is not necessary to achieve safety. A centralized control policy has to deal with the dimension explosion when the number of agents grow, while a decentralized design can significantly improve the scalability to a large number of agents.

## 3.2 LEARNING FRAMEWORK

From Proposition 1, we know that if we can jointly learn the control policy $\pi_i(s_i, o_i)$ and control barrier function $h_i(s_i, o_i)$ such that the decentralized CBF conditions in (2) are satisfied, then the multi-agent system is guaranteed to be safe. Next we formulate the optimization objective for the joint learning of $\pi_i(s_i, o_i)$ and $h_i(s_i, o_i)$. Let $T \subset \mathbb{R}_+$ be the time interval and $\tau_i = \{s_i(t), o_i(t)\}_{t \in T}$ be a trajectory of state and observation of agent $i$. Let $\mathcal{T}_i$ be the set of all possible trajectories of agent $i$. Let $\mathcal{H}_i$ and $\mathcal{V}_i$ be the function classes of $h_i$ and $\pi_i$. Define the function $y_i : \mathcal{T}_i \times \mathcal{H}_i \times \mathcal{V}_i \mapsto \mathbb{R}$ as:

$$y_i(\tau_i, h_i, \pi_i) := \min \left\{ \inf_{\mathcal{X}_{i,0} \cap \tau_i} h_i(s_i, o_i), \ \inf_{\mathcal{X}_{i,d} \cap \tau_i} -h_i(s_i, o_i), \ \inf_{\mathcal{X}_{i,h} \cap \tau_i} (\dot{h}_i + \alpha(h_i)) \right\}. \quad (3)$$

The set $\mathcal{X}_{i,h} := \{(s_i, o_i) \mid h_i(s_i, o_i) \geq 0\}$. Notice that the third item on the right side of Equation (3) depends on both the control policy and CBF, since $\dot{h}_i = \nabla_{s_i} h_i \cdot f_i(s_i, u_i) + \nabla_{o_i} h_i \cdot \dot{o}_i(t), u_i = \pi_i(s_i, o_i)$. It is clear that if we can find $h_i$ and $\pi_i(s_i, o_i)$ such that $y_i(\tau_i, h_i, \pi_i) > 0$ for $\forall \tau_i \in \mathcal{T}_i$ and $\forall i$, then the conditions in (2) are satisfied. For each agent $i$, assume that we are given $z_i$ i.i.d trajectories $\{\tau_i^1, \tau_i^2, \cdots, \tau_i^{z_i}\}$ drawn from distribution $\mathcal{D}_i$ during training. We solve the objective:

$$\text{For all } i, \text{ find } h_i \in \mathcal{H}_i \text{ and } \pi_i \in \mathcal{V}_i, \quad \text{s.t.} \quad y_i(\tau_i^j, h_i, \pi_i) \geq \gamma, \ \forall j = 1, 2, \cdots z_i, \quad (4)$$

where $\gamma > 0$ is a margin for the satisfaction of the CBF condition in (2). Following standard results from statistical learning theory, it is possible to establish the generalization guarantees of the solution to (4) to unseen data drawn from $\mathcal{D}_i$. See a detailed statement of the results in Appendix B. The bound depends on the richness of the function classes, the margin $\gamma$, as well as the number of samples used in (4). However, we note that such a generalization bound is only with respect to the *open-loop* data drawn from $\mathcal{D}_i$, the training data distribution, not to the *closed-loop* data in the testing when the learned controller is deployed. It is known to be challenging to handle the distribution shift between the training and testing due to the closed-loop effect. See, e.g., a recent result along this line in the context of imitation learning (Tu et al., 2021). We leave a systematic treatment of this closed-loop generalization guarantee for learning CBF in our future work. Finally, we note that in our experiments, we solve (4) and update the controller in an iterative fashion: run the closed-loop system with a certain controller to sample training data online and formulate (4), then update the controller by the solution of (4). We run the system using the updated controller and re-generate new samples to solve for a new controller. At the steady stage of this iterative process, the training and testing distribution shift becomes negligible, which validates the use of the generalization bounds given in Proposition 3. See more details of this implementation in Section 4.1, and more discussion on this point in the **Remark** in Appendix B.

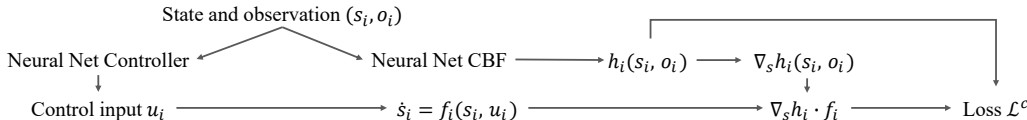

Figure 1: The computational graph of the control-certificate jointly learning framework in multi-agent systems. Only the graph for agent $i$ is shown because agents have the same graph and the computation is decentralized.

Besides the generalization guarantee, there also exists several other gaps between the formulation in (4) and the practical implementation. First, (4) does not provide a concrete way of designing loss functions to realize the optimization objectives. Second, there are still $N$ pairs of functions $(h_i, \pi_i)$ to be learned. Unfortunately, the dimension of the input $o_i$ of the functions $h_i, \pi_i$ are different for each agent $i$, and will even *change over time* in practice, as the proximity of other agents is time-varying, leading to time-varying local observations. To scale to an arbitrary number of agents, $h_i$ and $\pi_i$ should be invariant to the quantity and permutation of neighbourhood agents. Third, (4) does not provide ways to deal with scenarios where the decentralized CBF conditions are not (strictly) satisfied, i.e., where problem (4) is not even feasible, which may very likely occur when the system becomes too complex or the function classes are not rich enough. To this end, we propose effective approaches to solving these issues, facilitating the scalable learning of safe multi-agent control in practice, as to be introduced next.

## 4 SCALABLE LEARNING OF DECENTRALIZED CBF IN PRACTICE

Following the theory in Section 3, we consider the practical learning of safe multi-agent control with neural barrier certificates, i.e., using neural networks for $\mathcal{H}$ and $\mathcal{V}$. We will present the formulation of loss functions in Section 4.1, which corresponds to the objective in (4). Section 4.2 presents the neural network architecture of $h_i$ and $\pi_i$, which are invariant to the quantity and permutation of neighboring agents. Section 4.3 demonstrates a spontaneous policy refinement method that enables the control policy to satisfy the decentralized CBF conditions as possible as it could during testing.

### 4.1 LOSS FUNCTIONS OF JOINTLY LEARNING CONTROLLERS AND BARRIER CERTIFICATES

Based on Section 3.2. the main idea is to *jointly* learn the control policies and control barrier functions in multi-agent systems. During training, the CBFs regulate the control policies to satisfy the decentralized CBF conditions (2) so that the learned policies are safe. All agents are put into a single environment to generate experiences, which are combined to minimize the empirical loss function $\mathcal{L}^c = \Sigma_i \mathcal{L}^c_i$, where $\mathcal{L}^c_i$ is the loss function for agent $i$ formulated as:

$$\mathcal{L}^c_i(\theta_i, \omega_i) = \sum_{s_i \in \mathcal{X}_{i,0}} \max\left(0, \gamma - h^{\theta_i}_i(s_i, o_i)\right) + \sum_{s_i \in \mathcal{X}_{i,d}} \max\left(0, \gamma + h^{\theta_i}_i(s_i, o_i)\right)$$
$$+ \sum_{s_i \in \mathcal{X}_{i,h}} \max\left(0, \gamma - \nabla_{s_i} h^{\theta_i}_i \cdot f_i\left(s_i, \pi^{\omega_i}_i(s_i, o_i)\right) - \nabla_{o_i} h^{\theta_i}_i \cdot \dot{o}_i - \alpha(h^{\theta_i}_i)\right), \tag{7}$$

where $\gamma$ is the margin defined in Section 3.2. We choose $\gamma = 10^{-2}$ in implementation. $\theta_i$ and $\omega_i$ are neural network parameters. On the right side of Equation (7), the three items enforce the three CBF conditions respectively. Directly computing the third term could be challenging since we need to evaluate $\dot{o}_i$, which is the time derivative of the observation. Instead, we approximate $\dot{h}(s_i, o_i) = \nabla_{s_i} h^{\theta_i}_i \cdot f_i\left(s_i, \pi^{\omega_i}_i(s_i, o_i)\right) + \nabla_{o_i} h^{\theta_i}_i \cdot \dot{o}_i$ numerically by $\dot{h}(s_i, o_i) = [h(s_i(t+\Delta t), o_i(t+\Delta t)) - h(s_i(t), o_i(t))]/\Delta t$. For the class-$\mathcal{K}$ function $\alpha(\cdot)$, we simply choose a linear function $\alpha(h) = \lambda h$. Note that $\mathcal{L}^c$ mainly considers safety instead of goal reaching. To train a safe control policy $\pi_i(s_i, o_i)$ that can drive the agent to the goal state, we also minimize the distance between $u_i$ and $u^g_i$, where $u^g_i$ is the reference control input computed by classical approaches (e.g., LQR and PID controllers) to reach the goal. The goal reaching loss $\mathcal{L}^g = \Sigma_i \mathcal{L}^g_i$, where $\mathcal{L}^g_i$ is formulated as $\mathcal{L}^g_i(\omega_i) = \sum_{s_i \in \mathcal{X}} ||\pi^{\omega_i}_i(s_i, o_i) - u^g_i(s_i)||_2$. The final loss function $\mathcal{L} = \mathcal{L}^c + \eta \mathcal{L}^g$, where $\eta$ is a balance weight that is set to 0.1 in our experiments. We present the computational graph in Figure 1 to help understand the information flow.

In training, all agents are put into the specific environment, which is not necessarily the same as the testing environment, to collect state-observation pairs $(s_i, o_i)$ under their current policies with

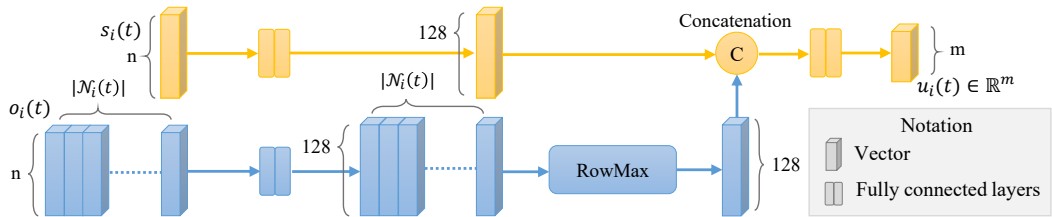

Figure 2: Neural network architecture of the control policy. The blue part indicates the quantity-permutation invariant observation encoder, which maps $o_i(t) \in \mathbb{R}^{n \times |\mathcal{N}_i(t)|}$ with time-varying dimension to a fixed length vector. The network takes the state $s_i$ and local observation $o_i$ as input to compute a control action $u_i$. The neural network of the decentralized CBF $h_i$ has a similar architecture except that the output is a scalar.

probability $1 - \iota$ and random policies with probability $\iota$, where $\iota$ is set to be $0.05$ in our experiment. The collected $(s_i, o_i)$ are stored as a temporary dataset and in every step of policy update, 128 $(s_i, o_i)$ are randomly sampled from the temporary dataset to calculate the total loss $\mathcal{L}$. We minimize $\mathcal{L}$ by applying stochastic gradient descent with learning rate $10^{-3}$ and weight decay $10^{-6}$ to $\theta_i$ and $\omega_i$, which are the parameters of the CBF and control policies. Note that the gradients are computed by back-propagation rather than policy gradients because $\mathcal{L}$ is differentiable w.r.t. $\theta_i$ and $\omega_i$.

**Iterative Data Collection and Training.** It is important to note that we did not use a fixed set of state-observation pairs to train the decentralized CBF and controllers. Instead, we adopted an on-policy training strategy, where the training data are collected by running the current system. The collected state-observation pairs are stored in temporary dataset that is used to calculate the loss terms and update the decentralized CBF and controllers via gradient descent. Then the updated controllers are used to run the system and re-generate new state-observation pairs as training data. The iterative data collection and training is performed until the loss converges. Such a training process is crucial for generalizing to testing scenarios. More discussion on this point can be found in the **Remark** in Appendix B.

### 4.2 QUANTITY-PERMUTATION INVARIANT OBSERVATION ENCODER

Recall that in Section 3.1, we define $o_i$ as the local observation of agent $i$. $o_i$ contains the states of neighboring agents and its dimension can change dynamically. In order to scale to an arbitrary number of agents, there are two pivotal principles of designing the neural network architectures of $h_i(s_i, o_i)$ and $\pi_i(s_i, o_i)$. First, the architecture should be able to dynamically adapt to the changing quantity of observed agents that affects the dimension of $o_i$. Second, the architecture should be invariant to the permutation of observed agents, which should not affect the output of $h_i$ or $\pi_i$. All these challenges arise from encoding the local observation $o_i$. Inspired by PointNet (Qi et al., 2017), we leverage the *max pooling* layer to build the quantity-permutation invariant observation encoder.

Let us start with a simple example with input observation $o_i(t) \in \mathbb{R}^{n \times |\mathcal{N}_i(t)|}$, where $n$ is the dimension of state and $\mathcal{N}_i(t)$ is the set of the neighboring agents at time $t$. $n$ is fixed while $\mathcal{N}_i(t)$ can change from time to time. The permutation of the columns of $o_i$ is also dynamic. Denote the weight matrix as $W \in \mathbb{R}^{p \times n}$ and the element-wise ReLU activation function as $\sigma(\cdot)$. Define the row-wise max pooling operation as $\mathrm{RowMax}(\cdot)$, which takes a matrix as input and outputs the maximum value of each row. Consider the following mapping $\rho : \mathbb{R}^{n \times |\mathcal{N}_i(t)|} \mapsto \mathbb{R}^p$ formulated as

$$\rho(o_i) = \mathrm{RowMax}(\sigma(W o_i)), \tag{8}$$

where $\rho$ maps a matrix $o_i$ whose column has dynamic dimension and permutation to a fixed length feature vector $\rho(o_i) \in \mathbb{R}^p$. The dimension of $\rho(o_i)$ remains the same even if the number of columns of $o_i(t)$, which is $|\mathcal{N}_i(t)|$, change over time. The network architecture of the control policy is shown in Figure 2, which uses the $\mathrm{RowMax}(\cdot)$ operation. The network of the control barrier function is similar except that the output is a scalar instead of a vector.

### 4.3 SPONTANEOUS ONLINE POLICY REFINEMENT

We propose a spontaneous online policy refinement approach that produces even safer control policies in testing than the neural network has actually learned during training. When the model dynamics or environment settings are too complex and exceed the capability of the control policy, the

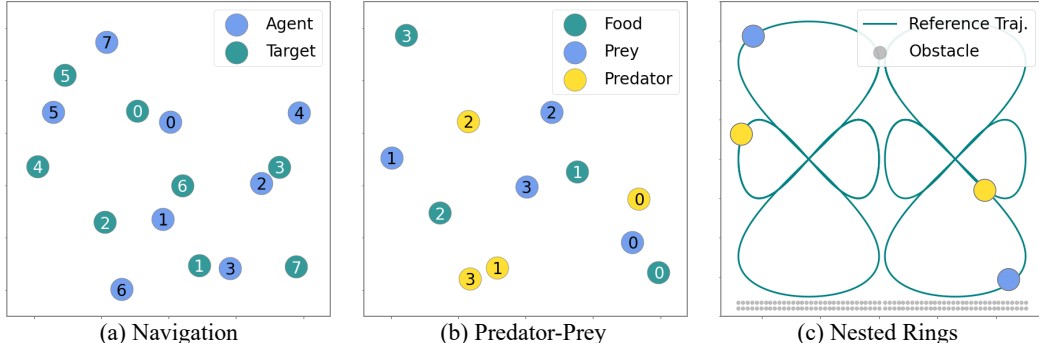

(a) Navigation         (b) Predator-Prey         (c) Nested Rings

Figure 3: Illustrations of the 2D environments used in the experiments. The *Navigation* and *Predator-Prey* environments are adopted from the multi-agent particle environment (Lowe et al., 2017). The *Nested-Rings* environment is adopted from Rodríguez-Seda et al. (2014).

decentralized CBF conditions can be violated at some points along the trajectories. Thanks to the control barrier function jointly learned with the control policy, we are able to refine the control input $u_i$ online by minimizing the violation of the decentralized CBF conditions. That is, the learned CBF can serve as a guidance on generating updated $u_i$ in unseen scenarios to guarantee safety. This is also a standard technique used in (non-learning) CBF control where the CBF $h$ is usually computed first using optimization methods like Sum-of-Squares, then the control inputs $u$ are computed online using $h$ by solving quadratic programming problems (Xu et al., 2017; Ames et al., 2017). In the experiments, we also study the effects of such an online policy refinement step.

Given the state $s_i$, local observation $o_i$, and action $u_i$ computed by the control policy, consider the scenario where the third CBF condition is violated, which means $\nabla_{s_i} h_i \cdot f_i(s_i, u_i) + \nabla_{o_i} h_i \cdot \dot{o}_i + \alpha(h_i) < 0$ when $h_i \geq 0$. Let $e_i \in \mathbb{R}^m$ be an increment of the action $u_i$. Define $\phi(e_i) : \mathbb{R}^m \mapsto \mathbb{R}$ as

$$\phi(e_i) = \max(0, -\nabla_{s_i} h_i \cdot f_i(s_i, u_i + e_i) - \nabla_{o_i} h_i \cdot \dot{o}_i - \alpha(h_i)) + \mu \|e_i\|_2^2. \tag{9}$$

If the first term on the right side of Equation (9) is 0, then the third CBF condition is satisfied. We can enforce the satisfaction in every timestep of testing (after $u_i$ is computed by the neural network controller) by finding an $e_i$ that minimizes $\phi(e_i)$. $\mu$ is a regularization factor that punishes large $e_i$. We set $\mu = 1$ in implementation and observed that in our experiment, a fixed $\mu$ is sufficient to make sure the $\|u_i + e_i\|$ do not exceed the constraint on control input bound. When evaluating on new scenarios and the constraints is violated, one can dynamically increase $\mu$ to strengthen the penalty. For every timestep during testing, we initialize $e_i$ to zero and check the value of $\phi(e_i)$. $\phi(e_i) > 0$ indicates that the control policy is not good enough to satisfy the decentralized CBF conditions. Then we iteratively refine $e_i$ by $e_i = e_i - \nabla_e \phi(e_i)$ until $\phi(e_i) - \mu \|e_i\|_2^2 = 0$ or the maximum allowed iteration is exceeded. The final control input is $u_i = u_i + e_i$. Such a refinement can flexibly refine the control input to satisfy the decentralized CBF conditions as much as possible.

## 5 EXPERIMENTAL RESULTS

**Baseline Approaches.** The baseline approaches we compare with include: MAMPS (Zhang & Bastani, 2019), PIC (Liu et al., 2020) and MADDPG (Lowe et al., 2017). For the drone tasks, we also compare with model-based planning method S2M2 (Chen et al., 2021). A brief description of each method is as follows. MAMPS leverages the model dynamics to iteratively switch to safe control policies when the learned policies are unsafe. PIC proposes the permutation-invariant critic to enhance the performance of multi-agent RL. We incorporate the safety reward to its reward function and denote this safe version of PIC as PIC-Safe. The safety reward is -1 when the agent enters the dangerous set. MADDPG is a pioneering work on multi-agent RL, and MADDPG-Safe is obtained by adding the safety reward to the reward function that is similar to PIC-Safe. S2M2 is a state-of-the-art model-based multi-agent safe motion planner. When directly planning all agents fails, S2M2 evenly divides the agent group to smaller partitions for replanning until paths that are collision-free for each partition are found. The agents then follow the generated paths using PID or LQR controllers.

For each task, the environment model is the same for all the methods. The exact model dynamics are visible to model-based methods including MAMPS, S2M2 and our methods, and invisible to the

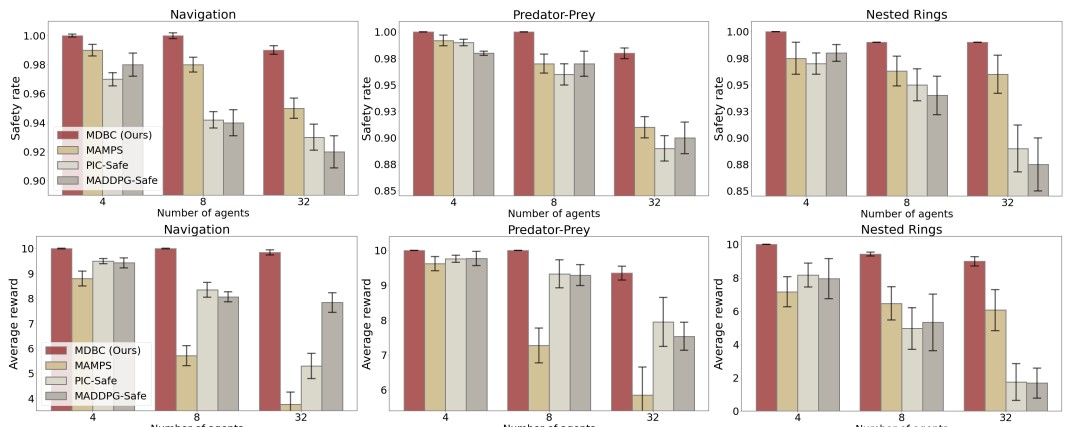

Figure 4: Safety rate and reward in the 2D tasks. Results are taken after each method converged and are averaged over 10 independent trials.

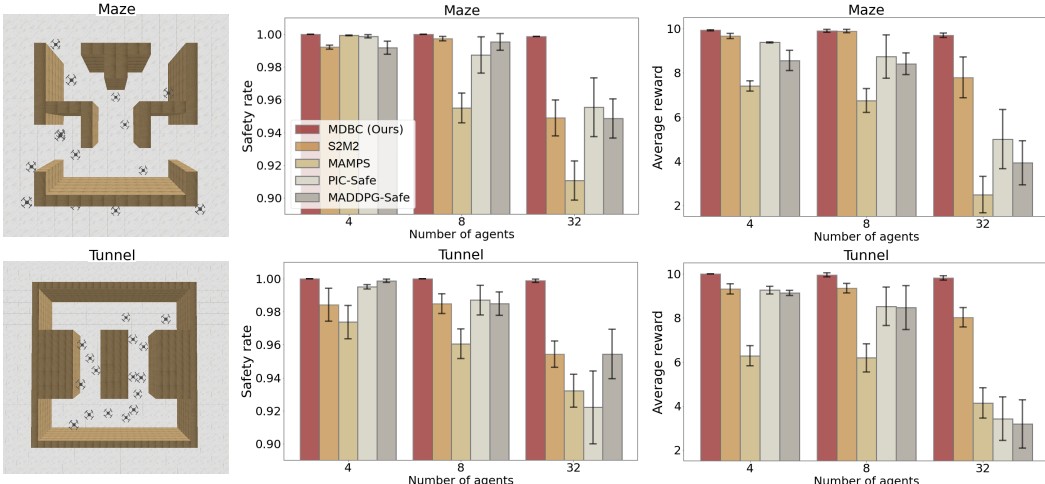

Figure 5: Environments and results of 3D tasks. In Maze and Tunnel, the initial and target locations of each drone are randomly chosen. The drones start from the initial locations and aim to reach the targets without collision. The results are taken after each method converged and are averaged over 10 independent trials.

model-free MADDPG and PIC. Since the model-free methods do not have access to model dynamics but instead the simulators, they are more data-demanding. The number of state-observation pairs to train MADDPG and PIC is $10^3$ times more than that of model-based learning methods to make sure they converge to their best performance. When training the RL-based methods, the control action computed by LQR for goal-reaching is also fed to the agent as one of the inputs to the actor network. So the RL agents can learn to use LQR as a reference for goal-reaching.

**Evaluation Criteria.** Since the primal focus of this paper is the safety of multi-agent systems, we use the *safety rate* as a criteria when evaluating the methods. The safety rate is calculated as $\frac{1}{N}\Sigma_{i=1}^{N}\mathbb{E}_{t\in T}\left[\mathbb{I}((s_i(t), o_i(t)) \in \mathcal{X}_s)\right]$ where $\mathbb{I}(\cdot)$ is the indicator function that is 1 when its argument is true or 0 otherwise. The observation $o_i$ contains the states of other agents within the observation radius, which is 10 times the safe distance. The safe distance is set to be the diagonal length of the bounding box of the agent. In addition to the safety rate, we also calculate the *average reward* that considers how good the task is accomplished. The agent is given a +10 reward if it reaches the goal and a -1 reward if it enters the dangerous set. Note that the agent might enter the dangerous set for many times before reaching the goal. The upper-bound of the total reward for an agent is +10, which is attained when the agent successfully reaches the goal and always stays in the safe set.

**Ground Robots.** We consider three tasks illustrated in Figure 3. In the Navigation task, each agent starts from a random location and aims to reach a random goal. In the Predator-Prey task, the preys aim to gather the food while avoid being caught by the predators chasing the preys. We only consider the safety of preys but not predators. In the Nested-Rings task, the agents aim to follow

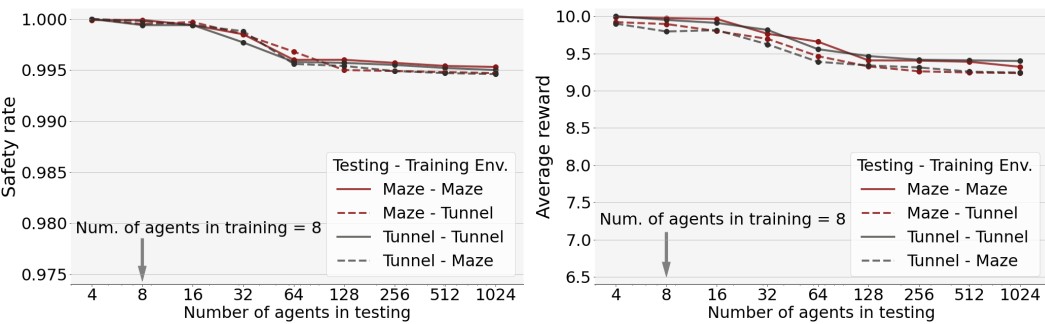

Figure 6: Generalization capability of MDBC in the 3D tasks. MDBC can be trained with 8 agent in one environment and generalize to 1024 agents in another environment in testing.

the reference trajectories while avoid collision. In order for the RL-based agents to follow the rings trajectory, we also give the agents a negative reward proportional to the distance to the nearest point on the rings. When adding more agents to an environment, we will also enlarge the area of the environment to ensure the overall density of agents remains similar.

Figure 4 demonstrates that when the number of agents grows (e.g., 32 agents), our approach (MDBC) can still maintain a high safety rate and average reward, while other methods have much worse performance. However, our method still cannot guarantee that the agents are $100\%$ safe. The failure is mainly because we cannot make sure the decentralized CBF conditions are satisfied for every state-observation pair in testing even if they are satisfied on all training samples due to the generalization error. We also show the generalization capability of MDBC with up to 1024 in the appendix and also visualization results in the supplementary materials.

**Drones.** We experiment with 3D drones whose dynamics are even more complex. Figure 5 demonstrates the environments and the results of each approach. Similar to the results of ground robots, when there are a large number of agents (e.g., 32 agents), our method can still maintain a high reward and safety rate, while other methods have worse performance. Figure 6 shows the generalization capability of our method across different environments and number of agents. For each experiment, we train 8 agents during training, but test with up to 1024 agents. The extra agents are added by copying the neural network parameters of the trained 8 agents. Results show that our method has remarkable generalization capability to diverse

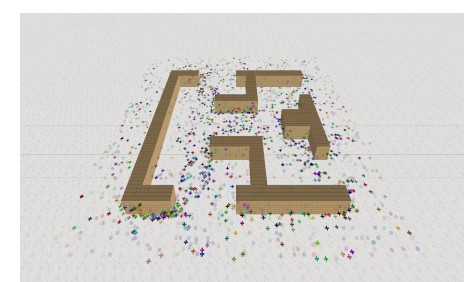

Figure 7: Illustration of the Maze environment with 1024 drones. Videos can be found in the supplementary material.

scenarios. Another related work Chen et al. (2020) can also handle the safe multi-drone control problem via CBF, but their CBF is handcrafted and based on quadratic programming to solve the $u_i$. Their paper only reported the results on two agents, and for 32 agents it would take more than 70 hours for a single run of evaluation (7000 steps and 36 seconds per step). By contrast, our method only takes $\sim 200s$ for a single run of evaluation with 32 agents, showing a significant advantage in computational efficiency. For both the ground robot and drone experiments, we provide video demonstrations in the supplementary material.

## 6 CONCLUSION

This paper presents a novel approach of learning safe multi-agent control via jointly learning the decentralized control barrier functions as safety certificates. We provide the theoretical generalization bound, as well as the effective techniques to realize the learning framework in practice. Experiments show that our method significantly outperforms previous methods by being able to scale to an arbitrary number of agents, and demonstrates remarkable generalization capabilities to unseen and complex multi-agent environments.

## 7 ACKNOWLEDGEMENT

The authors would like to thank Nikolai Matni for the valuable discussions. The authors acknowledge support from the DARPA Assured Autonomy under contract FA8750-19-C-0089. The views, opinions and/or findings expressed are those of the authors and should not be interpreted as representing the official views or policies of the Department of Defense or the U.S. Government.

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

## A  PROOF OF PROPOSITION 1

Since $\dot{h}_i = \nabla_{s_i} h_i \cdot f_i(s_i, u_i) + \nabla_{o_i} h_i \cdot \dot{o}_i(t)$, the satisfaction of (2) implies:

$$
\begin{aligned}
\forall\, (s_i, o_i) \in \mathcal{X}_{i,0}, & \qquad h_i(s_i, o_i) \geq 0 \\
\forall\, (s_i, o_i) \in \mathcal{X}_{i,d}, & \qquad h_i(s_i, o_i) < 0 \\
\forall\, (s_i, o_i) \in \{(s_i, o_i) \mid h_i(s_i, o_i) \geq 0\}, & \quad \dot{h}_i + \alpha\,(h_i) \geq 0.
\end{aligned}
\tag{10}
$$

The initial condition $(s_i(0), o_i(0)) \in \{(s_i, o_i) \mid h_i(s_i, o_i) \geq 0\}$ means that $h_i \geq 0$ at time $t = 0$. Since $\dot{h}_i + \alpha\,(h_i) \geq 0$, $h_i$ will stay non-negative, which is proved in Section 2 of Ames et al. (2014). This means that $(s_i(t), o_i(t)) \notin \mathcal{X}_{i,d}$ for $\forall t > 0$. Thus for $\forall i$ and $\forall t > 0$, agent $i$ would not enter the dangerous set, and the whole multi-agent system is safe by Definition 1. $\qquad\square$

**Remark.** Since the input dimension and permutation of $h_i$ can change with time, the time derivative of $h_i$ does not exist everywhere but almost everywhere. In fact, in the safety guarantee provided by Proposition 1, we do not require the time derivative of $h_i$ to exist everywhere. The $h_i$ can also be non-smooth. Based on (2) of Glotfelter et al. (2017), we can define $\dot{h}_i$ as the generalized gradient that always exists when $h_i$ is non-smooth and the time derivative exists almost everywhere. Then based on Theorem 2 of Glotfelter et al. (2017), as long as the CBF conditions are satisfied under the generalized gradient, then $h_i$ is a valid CBF and the safety can be guaranteed.

**Global CBF from decentralized CBFs.**   In addition to Proposition 1, another way to prove the global safety of the multi-agent system under the decentralized CBFs is to construct the global CBF $h_g : \mathcal{S} \mapsto \mathbb{R}$ from individual CBFs by taking the minimum, as is in:

$$
h_g(s) := \min\{h_1(s \downarrow s_1, s \downarrow o_1), h_2(s \downarrow s_2, s \downarrow o_2), \cdots, h_N(s \downarrow s_N, s \downarrow o_N)\},
\tag{11}
$$

where $s \downarrow s_i$ is the projection of the global state onto the state of agent $i$ and $s \downarrow o_i$ is the projection of the global state to the observation of agent $i$. Then the following proposition guarantees the global safety of the multi-agent system.

**Proposition 2.**  *If (10) is satisfied for every agent $i$, then the global CBF $h_g(s)$ satisfies:*

$$
\begin{aligned}
\forall\, s \in \mathcal{S}_0, & \qquad h_g(s) \geq 0 \\
\forall\, s \in \mathcal{S}_d, & \qquad h_g(s) < 0 \\
\forall\, s \in \{s \mid h_g(s) \geq 0\}, & \quad \dot{h}_g + \alpha\,(h_g) \geq 0,
\end{aligned}
\tag{12}
$$

where $\mathcal{S}_0 = \{s \in \mathcal{S} \mid \forall i, (s \downarrow s_i, s \downarrow o_i) \in \mathcal{X}_{i,0}\}$ and $\mathcal{S}_d = \{s \in \mathcal{S} \mid \exists i, (s \downarrow s_i, s \downarrow o_i) \in \mathcal{X}_{i,d}\}$. Then $\forall t > 0, h_g(s(t)) \geq 0$ and $s \notin \mathcal{S}_d$, which means the multi-agent system is globally safe.

*Proof.* Let us first prove that the satisfaction of (10) implies the satisfaction of (12). By definition of $\mathcal{S}_0$, when $s \in \mathcal{S}_0$, we have $\forall i, (s \downarrow s_i, s \downarrow o_i) \in \mathcal{X}_{i,0}$, which means $\forall i, h_i(s \downarrow s_i, s \downarrow o_i) \geq 0$. Thus $h_g(s) = \min_i\{h_i(s \downarrow s_i, s \downarrow o_i)\} \geq 0$. When $s \in \mathcal{S}_d$, we have $\exists i, (s \downarrow s_i, s \downarrow o_i) \in \mathcal{X}_{i,d}$, which means $\exists i, h_i(s \downarrow s_i, s \downarrow o_i) < 0$. Thus $h_g(s) = \min_i\{h_i(s \downarrow s_i, s \downarrow o_i)\} < 0$. When $s \in \{s \mid h_g(s) \geq 0\}$, we have $\forall i, (s \downarrow s_i, s \downarrow o_i) \in \{(s \downarrow s_i, s \downarrow o_i) \mid h_i(s \downarrow s_i, s \downarrow o_i) \geq 0\}$. So $\forall i, \dot{h}_i + \alpha\,(h_i) \geq 0$. Let $i^* = \arg\min_i\{h_i(s \downarrow s_i, s \downarrow o_i)\}$. Then $h_g(s) = h_{i^*}(s \downarrow s_{i^*}, s \downarrow o_{i^*})$ and $\dot{h}_g + \alpha\,(h_g) = \dot{h}_{i^*} + \alpha\,(h_{i^*}) \geq 0$. Hence the satisfaction of (10) implies the satisfaction of (12). Then based on Section 2 of Ames et al. (2014), we have $h_g(s(t)) \geq 0, \forall t > 0$. This means $s(t) \notin \mathcal{S}_d, \forall t > 0$, and the multi-agent system is globally safe. $\qquad\square$

## B  GENERALIZATION ERROR BOUND OF THE DECENTRALIZED CBF

To answer how well the learned $\pi_i(s_i, o_i)$ and $h_i(s_i, o_i)$ can generalize to unseen scenarios, we will provide a generalization bound with probabilistic guarantees. We denote the solution to (4) as $\hat{h}_i$ and $\hat{\pi}_i$. Denote the Rademacher complexity of the function class of $y_i$ as $\mathcal{R}_{z_i}(\mathcal{Y}_i)$, whose definition could be found in Appendix B. Also we define $\epsilon_i$ as the probability that the decentralized CBF conditions are violated for agent $i$ over randomly sampled trajectories (not necessarily the samples encountered in training). Under such definition, $\epsilon_i$ measures the *generalization error* and

can be expressed as $\epsilon_i = \mathbb{P}_{\tau_i \sim \mathcal{D}_i}\left[y_i(\tau_i, \hat{h}_i, \hat{u}_i) \leq 0\right]$. Then we have Proposition 3 that provides generalization guarantees for all the learned $\hat{h}_i$ and $\hat{\pi}_i$.

**Proposition 3** (Generalization Error Bound of Learning Decentralized CBF). *Assume that $|y| \leq b$ and (4) is feasible. Let $\hat{h}_i$ and $\hat{u}_i$ be the solutions to (4) and $\mu$ be a universal positive constant vector. Recall that $N$ is the number of agents. Then, for any $\delta \in (0, 1)$ the following statement holds:*

$$\mathbb{P}\left[\bigcap_{i=1}^{N}\left(\epsilon_i \leq \mu_i \frac{\log^3 z_i}{\gamma^2}\mathcal{R}_{z_i}^2(\mathcal{Y}_i) + \mu_i \frac{\log(N\log(4b/\gamma)/\delta)}{z_i}\right)\right] \geq 1 - \delta. \tag{6}$$

*Proof.* Note that $\epsilon_i = \mathbb{P}_{\tau_i \sim \mathcal{D}_i}\left[y_i(\tau_i, \hat{h}_i, \hat{u}_i) \leq 0\right] = \mathbb{E}_{\tau_i \sim \mathcal{D}_i}\left[\mathbb{I}\left(y_i(\tau_i, \hat{h}_i, \hat{u}_i) \leq 0\right)\right]$. Under zero empirical loss, using the Theorem 5 in Srebro et al. (2010), for any $\frac{\delta}{N} > 0$, the following statement holds with probability at least $1 - \frac{\delta}{N}$:

$$\epsilon_i \leq \mu_i \frac{\log^3 z_i}{\gamma^2}\mathcal{R}_{z_i}^2(\mathcal{Y}_i) + \mu_i \frac{\log(N\log(4b/\gamma)/\delta)}{z_i} \tag{13}$$

where $\mu_i > 0$ is some universal constant. By taking the union bound over all $N$ agents, the following statement holds with probability at least $(1 - \frac{\delta}{N})^N$:

$$\bigcap_{i=1}^{N}\left(\epsilon_i \leq \mu_i \frac{\log^3 z_i}{\gamma^2}\mathcal{R}_{z_i}^2(\mathcal{Y}_i) + \mu_i \frac{\log(N\log(4b/\gamma)/\delta)}{z_i}\right). \tag{14}$$

Since $(1 - \frac{\delta}{N})^N > 1 - \delta$ for $\delta \in (0, 1)$, we have:

$$\mathbb{P}\left[\bigcap_{i=1}^{N}\left(\epsilon_i \leq \mu_i \frac{\log^3 z_i}{\gamma^2}\mathcal{R}_{z_i}^2(\mathcal{Y}_i) + \mu_i \frac{\log(N\log(4b/\gamma)/\delta)}{z_i}\right)\right] \geq 1 - \delta, \tag{15}$$

which completes the proof. $\qquad\square$

The Rademacher complexity $\mathcal{R}_{z_i}(\mathcal{Y}_i)$ is defined as:

$$\mathcal{R}_{z_i}(\mathcal{Y}_i) := \sup_{\tau_i^1, \cdots \tau_i^{z_i} \sim \mathcal{D}_i} \mathbb{E}_{\xi \sim \text{Unif}\,(\{\pm 1\}^{z_i})} \sup_{h_i \in \mathcal{H}_i, \pi_i \in \mathcal{V}_i} \frac{1}{z_i}\left|\sum_{j=1}^{z_i} \xi_j y_i(\tau_i^j, h_i, \pi_i)\right|,$$

where $\xi \in \mathbb{R}^{z_i}$ is a random vector and $\xi_j$ denotes its $j^{th}$ element. $\mathcal{R}_{z_i}(\mathcal{Y}_i)$ characterizes the richness of function class $\mathcal{Y}_i$.

The left side of Equation (6) is the probability that the generalization error $\epsilon_i$ is upper bounded for all the $N$ agents. Equation (6) claims that the generalization error is bounded for all agents with high probability $1 - \delta$. Similar to the discussions in Section 4 in Boffi et al. (2020), for specific function classes of $\mathcal{H}_i$ and $\mathcal{V}_i$, such as Lipschitz parametric function or Reproducing kernel Hilbert space function classes, the Rademacher complexity of the function classes can be further bounded, leading to vanishing generalization errors as the number of samples $z_i$ increases. Such derivations are standard, and are thus omitted as they are not the focus of the present paper.

**Remark.** The generalization guarantee in Proposition 3 requires that the testing and training trajectories are drawn from the same distribution. Since in testing the trajectories come from the closed-loop (controller-in-the-loop) system, we should ensure that the training trajectories are also from the closed-loop system. Thus, in our implementation, the training data are not uniformly sampled from the state-observation space. Instead, the samples are drawn online under the current control policy, which is a solution to (4) using previously learned controller. We then use the updated controller to sample new data to formulate (4), and solve for an updated controller accordingly. In the experiments, we iterate this process until it converges. This way, at the steady stage of this process, the training and testing distribution shift becomes almost negligible, and the generalization results in Proposition 3 can thus be used. We leave a systematic analysis on the generalization and convergence of this iterative process in our future work.

## C  MODEL DYNAMICS

In the experiment section of our main paper, we use the 2D ground robots and 3D drones. For ground robots, we use a double integrator model with state $s_i = [x_i, y_i, v_{x,i}, v_{y,i}]$ for the navigation and predator-prey tasks, and the model from Rodríguez-Seda et al. (2014) for the nested rings task. For drones, we use the following dynamics:

$$s_i = \begin{bmatrix} x_i \\ y_i \\ z_i \\ v_{x,i} \\ v_{y,i} \\ v_{z,i} \\ \theta_{x,i} \\ \theta_{y,i} \end{bmatrix}, \quad \frac{ds_i}{dt} = \begin{bmatrix} v_{x,i} \\ v_{y,i} \\ v_{z,i} \\ g\, tan(\theta_{x,i}) \\ g\, tan(\theta_{y,i}) \\ a_{z,i} \\ \omega_{x,i} \\ \omega_{y,i} \end{bmatrix}, \quad u_i = \begin{bmatrix} \omega_{x,i} \\ \omega_{y,i} \\ a_{z,i} \end{bmatrix}. \tag{16}$$

## D  SUPPLEMENTARY EXPERIMENT

For 3D drones we have shown the generalization capability to 1024 agents even when our method is trained with 8 agents (Figure 6). For 2D ground robots we have similar results that were omitted in the main paper due to space limitations. We present the results in Figure 8 as below. Our method demonstrates the exceptional generalization capability to testing scenarios where the number of agents is significantly greater than that in training. The safety rate and average reward remain high even when the number of agents grow exponentially.

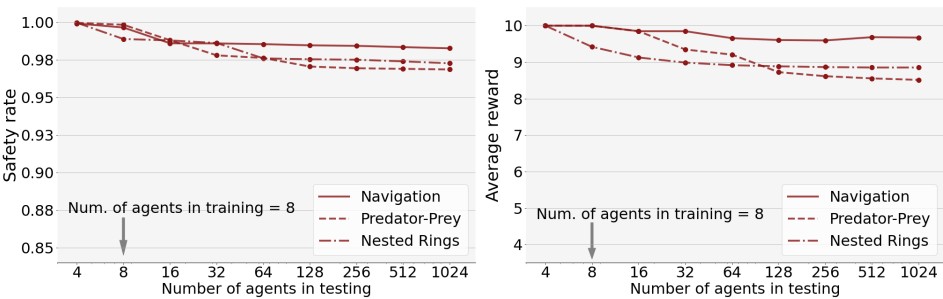

Figure 8: Generalization capability of our method in the 2D tasks. Our method is trained with 8 agents and tested with up to 1024 agents.

**Ablation Study on Online Policy Refinement.**  In Section 4.3, we introduced a test-time policy refinement method. Here we study the effect of this method on our performance and present the results in Table 1. It is shown that even without the OPR, the safety rate is still promising. The OPR further improved the safety rate. The steps requiring OPR in testing only accounts for a small proportion ($< 17\%$) of the total steps. The proportion gradually becomes saturated and does not significantly increase as the number of agents grow.

Table 1: Effect of online policy refinement (OPR). Proportion of OPR stands for the proportion of steps that OPR is performed in testing.

| Environment | Config | Safety Rate | | | | Proportion of OPR | | | |
| --- | --- | --- | --- | --- | --- | --- | --- | --- | --- |
| | | 4 Agents | 8 Agents | 32 Agents | 1024 Agents | 4 Agents | 8 Agents | 32 Agents | 1024 Agents |
| Maze | w/ OPR | 0.9999 | 0.9999 | 0.9987 | 0.9956 | 0.0149 | 0.0958 | 0.1423 | 0.1655 |
| | w/o OPR | 0.9999 | 0.9999 | 0.9869 | 0.9741 | 0 | 0 | 0 | 0 |
| Tunnel | w/ OPR | 0.9999 | 0.9998 | 0.9988 | 0.9946 | 0.0117 | 0.0729 | 0.1271 | 0.1493 |
| | w/o OPR | 0.9999 | 0.9992 | 0.9866 | 0.9727 | 0 | 0 | 0 | 0 |

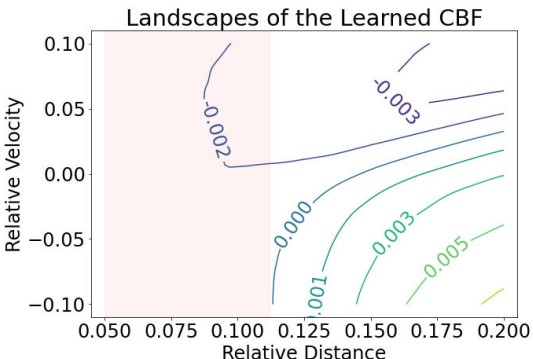

Figure 9: Visualization of the learned CBF in the Maze environment with 2 agents. The red area is where the distance between agents is less that the safe threshold.

**Visualization of the Learned CBF.**    To have a better understanding of the learned decentralized CBF, we provide a visualization in Figure 9. The CBF is learned in the Maze environment with two agents, in order to simplify the interpretation. The relative distance is defined in the 3D Euclidean space. The relative velocity is the negative time derivative of the relative distance. When the relative velocity is positive, the two agents are getting close to each other. From Figure 9, we know that the learned CBF is negative on the dangerous states (the red area) and the potentially dangerous states (the top-right area), where the agents are moving towards each other. The CBF is positive only when the states are sufficiently safe (the bottom-right area).

