# OpenReview forum: "Learning Safe Multi-agent Control with Decentralized Neural Barrier Certificates"
_ICLR.cc/2021/Conference — ICLR 2021 Poster_

### Official Review · AnonReviewer3 · 2020-10-17
**Interesting idea and results but several issues**

**Rating:** 4
**Confidence:** 4

**Review:**

Summary:
This paper presents a learning framework for decentralized control barrier functions in multi-agent systems, applied to collision-avoidance problems. The paper reviews some key ideas from the CBF literature and extends them to the multi-agent, partially-observed case, then presents a learning framework for estimating CBFs and policies jointly, and concludes with experimental results.

Strengths:
* Good coverage of recent literature in this area and placement of the present work within it
* Adequate, though concise presentation of key ideas from CBF literature. I was able to follow this fine but (a small suggestion) it might be a good idea to spend a little more time on this for readers who have not heard of CBFs before.
* Clear extension to the decentralized, multi-agent case
* Strong experimental evidence that this approach can scale

Weaknesses:
* (main one) the time derivative of each agent's observation may not always exist - indeed the paper points out that observations can arbitrarily permute and change dimension at any time. This would seem to be a pretty big theoretical problem...
* the paper should comment explicitly upon how this formulation differs from existing work in (non-learned) multi-agent CBFs
* the interpretation of (3) is unclear. One component (though it would still be unclear even without this issue), is the parsing of \mathcal{X}_{i, 0}. Is this _all state, observation pairs for agent i_? If so, it would certainly help to just say that.
* The presentation and preparation for Prop 2 is very sparse and hard to follow/understand. For example, what does (5) really buy us except for a term in (6) which is not really explained clearly? To me, this looks a little bit "decorative" (to use a term I've seen several conferences use in instructions to reviewers).
* the mu term in (9) doesn't really make sense
* in the results, it would be good to comment on failure modes since the proposed method does not achieve perfect safety rates and in the video it is clear that some of the agents violate safety
* Is MAFACTEST published? If not, this would seem to merit further discussion. If so, it should be clearly cited as different from FACTEST.

Overall:
I like the main ideas here, but I have identified a number of directions of weakness and potential improvement. For now, I cannot recommend publication.

---

> ### Author Response · Authors · 2020-11-16
> **Response to Reviewer 3**
>
> Thanks for finding our paper well-written and experimental results strong. The points you raised are explained in the following. We have provided an updated paper and code as part of our response.
>
> **Q1: The time derivative of each agent's observation may not always exist, because the dimension and permutation can change over time.**
>
> A1: Although the dimension and permutation can change, the observation will become a *fixed-length* vector after passing through our *quantity-permutation invariant* observation encoder. This was in fact one of the significant advantages of our approach, by using the quantity-permutation invariant neural network architectures. The time derivative of the fixed-length vector $v$ then exists, and can be estimated numerically by $[v(t+\Delta t) - v(t)]/\Delta t$. Thus we do not need to explicitly deal with the time derivative of the observation $o_i$. We also updated Section 4.1 to make it clear on how to avoid direct computation of $\dot{o}_i$.
>
> **Q2: How does this formulation differ from existing work in (non-learned) multi-agent CBFs?**
>
> A2: The majority of the multi-agent CBF works use a *centralized* CBF. A few very recent works like  <Guaranteed Obstacle Avoidance for Multi-Robot Operations With Limited Actuation: A Control Barrier Function Approach> introduced *decentralized CBF*. However, these are all  non-learning based approaches, and the decentralized CBF is hand-crafted, which is very hard to construct for complex systems (please also see our response to Reviewer 4 Question 3). The main contribution of this paper is *NOT* the formulation of multi-agent CBFs, which is close to some of the non-learning multi-agent CBF in literature. Instead, our contribution is to propose the first *fully decentralized multi-agent CBF learning framework that can accommodate* a practical  *joint learning* of safe multi-agent control policies and CBF certificates.
>
> **Q3: The interpretation of (3) is unclear.**
>
> A3: The three $\inf$ terms on the right-hand side correspond to the three CBF conditions in (2). If any of the $\inf$ terms is negative (i.e., the minimum of the three terms is negative), then the CBF conditions are violated. The $y_i$ function takes the minimum of the three terms over all state-observation pairs on the trajectory $\tau_i$. Any $h_i$ (with corresponding $\pi_i$) that makes $y_i$ in (3)  positive is a valid CBF and therefore, $y_i$ unifies the 3 CBF conditions in (2). In this way, $y_i$ can be used to design the empirical loss as in Equation (7). Any global optimizer of (7) gives a valid CBF. $\mathcal{X}_{i, 0}$ denotes all possible initial state-observation pairs for agent $i$.
>
> **Q4: Proposition 2 is hard to follow.**
>
> A4: The interpretation of Proposition 2 is not as hard as it seems to be. The proposition claims that the generalization error (i.e., $\epsilon_i$, which is the probability that the CBF condition is not satisfied in testing) is bounded for all agents with high probability (i.e., $1 - \delta$, as is in Equation (6)). Equation (5), which is the definition of the Rademacher complexity, measures the “richness” of the function class, which is a common terminology in statistical learning theory, and is indispensable in our generalization results in Proposition 2. Its importance has been recognized by other reviewers.
>
> **Q5: The $\mu$ in (9) does not make sense.**
>
> A5: We have added a detailed explanation in the highlighted part in Section 4.3. In short, $\mu$ is a regularization factor that punishes large $e_i$ to avoid large $u_i+e_i$, since in practice, the control systems normally have actuation limits (i.e. upper bounds for $|| u_i + e_i ||$ ) and therefore do not allow large $e_i$.
>
> **Q6: Discussion on the failure mode.**
>
> A6: The failure is mainly because we cannot guarantee that the decentralized CBF conditions are satisfied for every state-observation pair in *testing* even if the *training* samples satisfy the CBF conditions, due to the generalization error identified in Proposition 2. We have added the discussion in the experiment section (see the highlighted part of the *Ground Robot* paragraph). In the video the drones have very few collisions (less than 0.5% of the total number of steps) but it may appear to be more than the number of actual collisions because of the visualization angle.
>
> **Q7: Is MAFACTEST published?**
>
> A7: MAFACTEST is a multi-agent extension of FACTEST. It is still under anonymous review and will be published soon and we will add the citation at that time. We have uploaded the draft of the MAFACTEST paper in the supplementary material, where the name is changed to S2M2, for the reviewers’ reference.
>
> We thank the reviewer again for appreciating our idea, and sincerely hope that our explanation and updated draft with more detailed information have cleared all your concerns. We are looking forward to more discussions to further improve our paper.

---

> > ### Comment · AnonReviewer3 · 2020-11-18
> > **Thank you for your clarification**
> >
> > I appreciate your careful response to my review and clarifications of my many questions. My take is gravitating toward:
> > - The paper contribution of jointly learning control and barrier function in the decentralized context appears novel.
> > - I'm not sure I understand or believe any theoretical justification for why any fixed-length encoding scheme for the observations yields a safe control law. Am I missing something?
> > - Similarly, using a fixed \mu penalization for input saturation seems like a simple approximation to a penalty method in constrained optimization. I would be curious to know if this yielded constraint satisfaction (which I would guess might only happen in very special cases)? Otherwise, what can we really conclude about safety?
> > - The section dealing with Rademacher complexity still strikes me as practically unnecessary, in that I'm still not sure I understand the practical meaning of this section and the resulting generalization error bound. What does it imply about safety? Further discussion in the paper is certainly warranted.
> >
> > In light of your feedback and reading the other reviewers' comments, I am certainly less confident in my overall appraisal of the paper. However, I hesitate to change my score before discussing with the other reviewers and/or having the above questions answered.

---

> > > ### Author Response · Authors · 2020-11-19
> > > **Thanks for recognizing the novelty of our work**
> > >
> > > Thanks a lot for recognizing the novelty of our work. We hope the following response can address your remaining concerns.
> > >
> > > **Q1: Why does any fixed-length encoding yield a safe control law?**
> > >
> > > A1: The fixed-length encoding layer is a part of the $h_i$ as a specially structured neural network in order to deal with time-varying quantity and permutation of neighboring agents, which is one key challenge in multi-agent CBF learning. Without the fixed-length encoding, it could be difficult to implement $h_i$ in practice, since $h_i$ cannot dynamically change the size of its input layer to adapt to a different number of neighbouring agents. Therefore, the fixed-length encoding makes $h_i$ implementable in practice. However, please note that the input of $h_i$ is still the original state-observation pair, not the fixed-length encoding output. Hence, the theoretical guarantee of safety, which hinges on the CBF $h_i$ per se is not affected by this encoding layer, i.e., the statement in Section 3.1 still holds.
> > >
> > > **Q2: Using a fixed $\mu$ in (9) is similar to the penalty method in constrained optimization. Can this yield constraint satisfaction?**
> > >
> > > A2: Yes, fixing $\mu$ shares the similar idea as the penalty method in constrained optimization. We also agree that an inappropriate choice of $\mu$ will make the constraint violated. However, in our implementation, we use this penalty approach for the following reasons. First, we have actually tried to solve the “hard-constrained” problem instead at the beginning. However, it was not very computation efficient, and was not able to yield the online execution as we desired. With the penalty approach, we were able to solve the problem much faster and in an online fashion, which is one advantage of our current implementation. Second, in fact, in most of our experiments, a fixed $\mu$ already yields prominent performance without constraint violation, which was why we kept it fixed. Third, this $\mu$ essentially provides a knob to tune our algorithm: if the constraints are violated, one can increase the value of $\mu$. From the theory of constrained optimization, we know that this is essentially the update on the “dual” variable (Lagrange multiplier). In other words, we never intended to “fix” $\mu$, but instead intended to leave it tunable, which essentially falls back to solving the constrained optimization directly. Thanks again for pointing this out. We have updated the submission with a clarification on this point.
> > >
> > > **Q3: What is the practical meaning of Section 3.2 that deals with the Rademacher complexity and generalization error bound?**
> > >
> > > A3: In our paper, the CBFs are learned from data and the empirical loss is minimized over samples. There is a natural question on whether the learned CBFs can generalize to testing scenarios. Section 3.2 answers how well the CBFs can *generalize* to unseen samples in testing,  in that the *generalization error* is actually *bounded*, and quantifies how the bound depends on the number of samples, the richness of the function class used, and other problem parameters. We borrowed the ideas from statistical learning theory to derive the generalization error bound in (6) (which has been deemed to be sound and one of the strengths of our paper by some other reviewers). The Rademacher complexity is necessary in (6) to quantify the “richness” of the function class. We agree that the definition of Rademacher complexity (5) is a bit abstract and lack of practical meaning. However, theoretically, the bound in (6) still gives us the rationale about whether and how our learning approach ensures the CBF condition to be satisfied (which further ensures safety), when a *finite number of samples* and *certain function classes* are used in our algorithm.  As per your suggestion, we have moved the definition of Rademacher complexity to the appendix. In the latest version, we have also added a clarification of the purpose of Section 3.2 in its first paragraph.
> > >
> > > We sincerely thank the reviewer again for the timely and valuable comments. We hope that our response and revision, together with the strengths identified by other reviewers, have cleared most of your concerns, and can lead to a re-evaluation. We are also always open to any further suggestions that may help improve our paper in the final version.

---

> > > > ### Comment · AnonReviewer3 · 2020-11-19
> > > > **Still don't follow...**
> > > >
> > > > Apologies, but I still do not follow how h can even be well-defined. My fundamental question is: isn't the input to h (the observation) changing dimension and permutation all the time? I understand that a key contribution is passing the observation through a fixed-size-encoding network, so I see that it can at least be computed, but I'm not sure I see how it is possible to ensure that its time derivative always exists and that it always satisfies a CBF condition. I buy that it exists *almost everywhere* (except at points where the observation order/dimension changes). Is that enough? Either way, some discussion on this point is definitely warranted.
> > > >
> > > > In the meantime, thank you for addressing the other two questions above.

---

> > > > > ### Author Response · Authors · 2020-11-20
> > > > > **Response to the reviewer**
> > > > >
> > > > > Thank you for clarifying your concerns. We have updated the paper submission and also explained here why we only need $h_i$ to be differentiable “almost everywhere” instead of everywhere.
> > > > >
> > > > > In fact, it is known in the literature of CBF-based safety certificates that the time derivative of $h_i$ does not need to exist everywhere in order to make $h_i$ a CBF that guarantees safety. In [1], the authors have proved that $h_i$ can also be non-smooth for multi-agent systems. Based on (2) of [1], $\dot{h}_i$ can be defined as the generalized gradient that always exists when $h_i$ is non-smooth, and the time derivative exists almost everywhere. Then based on Theorem 2 of [1], as long as the CBF conditions are satisfied under the generalized gradient, then $h_i$ is a valid CBF and the safety can be guaranteed. Actually, the contribution of paper [1] is to prove that non-smooth barrier functions $h$ can still lead to safety guarantees. In [1], the non-smoothness is caused by having $max$ or $min$ functions, which is exactly our case due to the Max pooling layer in PointNet. To better clarify this point, we have updated Appendix A (see the Remark) to reflect this. We appreciate your comments that help improve both the clarity and completeness of our draft. Hope our explanation and the updated draft have cleared this only remaining concern.
> > > > >
> > > > > [1] Glotfelter, Paul, Jorge Cortés, and Magnus Egerstedt. "Nonsmooth barrier functions with applications to multi-robot systems." IEEE control systems letters 1.2 (2017): 310-315.

---

> > > > > ### Author Response · Authors · 2020-11-23
> > > > > **Looking forward to further discussions!**
> > > > >
> > > > > Dear reviewer,
> > > > >
> > > > > We were wondering if our last response has cleared your remaining concern. In the remaining 2 days of the rebuttal period, we would appreciate it if you could kindly let us know whether you have any other questions, so that we can still have time to respond and address. We look forward to your reply and are happy to have further discussions. Thanks!
> > > > >
> > > > > Best regards,
> > > > >
> > > > > The Authors

---

### Official Review · AnonReviewer1 · 2020-10-28
**Comments**

**Rating:** 6
**Confidence:** 3

**Review:**

This paper extends the neural barrier certificate method from reinforcement learning to  the decentralized multi-agent reinforcement learning setting. In particular,  it borrows the idea of the control barrier function, which enforces the states of the dynamic system to stay in the safe set. It is known that in the single agent system if the control barrier function h satisfies the equation (1), the agent will never enter the dangerous set under policy pi. A straightforward way to extend the control barrier function in the MARL setting is to replace the state-action pair by  the joint state-action pair  in equation (1). However it may suffer from the exponentially large state and action space. Therefore, the author proposes a decentralized setting, where each agent just maintains its own control barrier function. To learn the policy and function h simultaneously, the author penalizes the violation of equation (2). At last, they test the proposed method in Navigation, Predator-Prey, ground robots and Drones compare it with several baselines.


Concern 1: The contribution of this paper is incremental. Particularly, the author extends the control barrier function from (1) to (2). It is quite straightforward to prove proposition 1 and 2. In addition, I am not sure whether the definition of equation (2) is reasonable or not.  To simplify the problem, it directly assumes h_i are independent. Is it true that the whole barrier function isthe product of the individual one?

Concern 2: As the author mentioned, the loss function in equation (7) just considers the safety instead of goal reaching. Thus how do you train the agent in the experiment?  It is initialized by a well-trained policy (but not safe ), and then rectified by the loss function. Or you train the goal-seeking policy and safe-policy simultaneously.


Concern3: the definition of  safety of the Multi-agent system. It suggests that safety is defined on the Euclidean distance between agents (the agent does not collide with each other). Is it possible to extend this definition to a more general setting?

---

> ### Author Response · Authors · 2020-11-16
> **Response to Reviewer 1**
>
> Thanks for your valuable comments. We hope the following statement can address your concern. We have provided an updated paper and code as part of our response.
>
> **Q1: Is the contribution of this submission incremental?**
>
> A1: We respectfully disagree that our contribution is incremental for the following reasons: 1) Although the notions of CBF and multi-agent CBF exist in many previous papers, constructing CBF even for a single agent is known to be an open and extremely hard problem (see our related work section). Most previous works on CBF heavily rely on hand-crafted CBF and therefore only work for very simple dynamics. Learning CBF for a single agent is getting popular these days but to the best of our knowledge, there is no published work on *learning CBF* for multi-agent systems. 2) The extension from (1) to (2) is straightforward and we *had never* claimed that it is our main contribution. However, our *fully decentralized CBF* is the first well-defined decentralized CBF that can accomodate a  framework to *jointly learn* safe multi-agent control policies and CBF certificates. As identified by the other reviewers, this is definitely not a trivial step and we believe it is a fundamental breakthrough in multi-agent safe control with barrier certificates. 3) Our  techniques that make the learning framework more effective and scalable for practical multi-agent systems, especially  the use of *quantity-permutation invariant* neural network architectures in learning to handle the *permutation of neighboring agents* is the first to bring advanced neural network structure to learning for multi-agent safe control. 4) Our comprehensive experiments demonstrate that we significantly outperform both leading non-learning and learning  approaches. In the multi-agent safe control literature, there is no published work that can handle an arbitrary number of agents, or >1000 nonholonomic systems scalably and safely as we have shown in our experiments. The novelties of these contributions have now also been acknowledged by other reviewers.
>
> **Q2: Is the definition in (2) correct?**
>
> A2: The definition is correct and we have added more explanation in the updated paper in  Section 3.1. As long as the conditions in (2) are satisfied, the system is guaranteed to be safe, which is already formally stated in Proposition 1. This can also be justified by the paper <Guaranteed Obstacle Avoidance for Multi-Robot Operations With Limited Actuation: A Control Barrier Function Approach>. The individual $h_i$s are not independent in the sense that they depend on $o_i$, which is a function of other agents’ state. In the training process, since we assume homogeneous control policies and barrier certificates for all agents, the agents’ interaction is already taken into account. For agent $i$, the control policy not only affects how agent $i$ chooses $u_i$ to make its own barrier function $h_i$ satisfy the CBF condition, but also how its neighbors behave to help satisfying $h_i$ by influencing $o_i$. In the same manner, the behavior of agent $i$ also affects its neighbors’ barrier functions.
>
> If one is to construct one single overall barrier function, it is not the product of the individual ones but is the *minimum* of the barrier function for each agent, as is formulated in the highlighted part in Appendix A. When this global CBF is positive, all decentralized CBFs are positive and the agents are safe.
>
> **Q3: How are the agents trained in the environment?**
>
> A3: We apologize for the confusion. We have added a detailed description in the brown part in Section 4.1, including the loss function for goal-reaching and how the training is exactly performed. The safety loss and goal-reaching loss are minimized simultaneously. The policies are randomly initialized. Please feel free to let us know if any other information is needed.
>
> **Q4: Is it possible to extend the safety defined on Euclidean distance to a more general setting?**
>
> A4: Our definition of safety is not limited to the Euclidean distance (in terms of the collision occurring or not). Any safety condition defined on the *state-observation* space is applicable, e.g., the relative speed, the remaining battery charge if the charge is also a state variable. In general, as long as the safety constraints can be represented as a forward invariant set {x | h(x) > 0}, it can be guaranteed using our learning framework. We chose the Euclidean distance in the paper as it is straightforward to understand and demonstrate. We will involve other more general safety properties in our future work.
>
> We sincerely hope that our response has cleared the concern you had regarding our contribution, and we are looking forward to more discussions.

---

> ### Author Response · Authors · 2020-11-19
> **Safety guarantee from the global CBF**
>
> In addition to our [**previous response**](https://openreview.net/forum?id=P6_q1BRxY8Q&noteId=H_clP6U3A6l), we have also proved in **Appendix A (the highlighted part)** that the global safety of multi-agent control can be guaranteed by the global CBF, which is the minimum of all individual CBFs for every agent. We also proved in Appendix A (the highlighted part) that the satisfaction of individual decentralized CBF conditions implies the satisfaction of the global CBF conditions.
>
> We sincerely hope that the proof can clear your concern on whether the decentralized CBFs can effectively guarantee global safety. We hope that our response, our detailed revision of the submission as well as the novelty identified by other reviewers can lead to a re-evaluation. We look forward to your reply and are happy to have more discussion.

---

> ### Author Response · Authors · 2020-11-23
> **Looking forward to further discussions!**
>
> Dear reviewer,
>
> We were wondering if our response and revision have cleared all your concerns. In the previous 2 responses, we have tried to address all the 5 points you have raised. In the remaining 2 days of the rebuttal period, we would appreciate it if you could re-evaluate our submission, or kindly let us know whether you have any other questions, so that we can still have time to respond and address. We are looking forward to  discussions that can further improve our current manuscript. Thanks!
>
> Best regards,
>
> The Authors

---

> > ### Comment · AnonReviewer1 · 2020-11-25
> > **Re:Responds**
> >
> > Thanks for the responds. After read that, I raise my score to 6.

---

> > > ### Author Response · Authors · 2020-11-25
> > > **Thanks for raising the score!**
> > >
> > > We appreciate the reviewer for raising the score to 6! Thanks for the valuable comments and suggestions!

---

### Official Review · AnonReviewer4 · 2020-10-28
**An interesting paper that tackles multi-agent safety control via learning the control barrier by neural nets**

**Rating:** 8
**Confidence:** 4

**Review:**

##########################################################################
Post rebuttal:

I have read authors' rebuttal as well as the newly added contents.
I appreciate the time and the details that the author goes in when addressing my comments.
I am convinced that this method can be used even without the OPR module based on the ablation study.

Overall, I think this is a high quality paper that deals with decentralised Multi-agent safety-awared learning. And learning the safe certificate is the shining point of this contribution. I recommend for acceptance and give my final score at 8.


##########################################################################
Summary:

The work studies how, in a multi-agent environment, agents should avoid collisions and develop safety guarantee during learning. It presents a novel approach of learning safe multi-agent control via jointly learning the control barrier functions (CBF) as safety certificates. The benefit of this learning-based CBF framework is that during testing, it is decentralised, thus it can scale up to large number of agents, and is agnostic to the changing agent number, which nicely fits with the need of multi-agent problems. An extensive list of experiments against strong multi-agent learning baselines has been conducted, and results are significant.

##########################################################################
Reasons for score:

Extending the learning-based CBF approach to multi-agent problems is an important work in the area of multi-agent learning. The result of multi-agent safe control of robots and drones on 1024 agents using the policies trained on only 8 agents to me is impressive.  I am tending towards an acceptance. However,  I have some uncertainties over the the current version, see the below.


##########################################################################ProPros:

1. This paper studies an interesting and an emerging topic in multi-agent learning: multi-agent safety control. They used a CBF approach, a classical idea in control domain, and empower it with the learning capability and extend it in the multi-agent domain.

2. It gives out a clear definition of multi-agent safety control, and show that why a decentralised CBF can meet such multi-agent safety control definition.

3. The author provides a classical high probability generalisation bound based on the Rademacher complexity.

2. The author is considerate in that they account for the changing number of neighbours in a time-variant multi-agent setting. I believe it is an important topic but usually would be neglected by MARL people from the moment of problem definition.



3. This paper provides comprehensive experiments, including 2D grid world, ground robots, and 3D drones. The results seems competitive against baselines, and outperforming when the agent number is large.

##########################################################################
Cons:

1. I don't quite get the idea of section 4.3, the spontaneous online policy refinement. It seems to me that the learning agent can somehow “cheat” during the test time should they entered into a dangerous zone during the test time. I believe an ablation study on the performance with and without this module is critical.

2. Can the author provide more insights why the proposed algorithm outperform the baselines when the agent number is large, based on figure 4 and 6, it is consistent that N=32 is the break-even point.

3. The related work of <Guaranteed Obstacle Avoidance for Multi-Robot Operations With Limited Actuation: A Control Barrier Function Approach> seems to be doing a similar topic. Could the author include this baseline into the experiment?


4. the union bound in equation6, should it be a \cup rather than \cap?


5. am I right that your method is in fact a supervised learning approach to learn the policy and the certificate, rather than a RL approach, where the policy update will not influence the new data that you are going to collect. I would love to see a pseduco-code if it is not the case.

6. Can the author comment on how in general the dynamical system information of f_i can be known?

##########################################################################
Questions during rebuttal period:

Please address and clarify the cons above

#########################################################################

---

> ### Author Response · Authors · 2020-11-16
> **Response to Reviewer 4**
>
> Thanks for finding our results important and impressive. We have improved the current version as per your suggestions. All new information is highlighted in brown. We clarify our statements as follows.
>
> **Q1: More explanation on the test-time online policy refinement.**
>
> A1: The online policy refinement is not “cheating” but instead a standard technique used in control theory. In non-learning-based control, the CBF $h$ is usually computed first using optimization methods like Sum-of-Squares then the control inputs $u$ are computed online using $h$ by solving quadratic programming problems. Technically any $u$ that makes $h$ satisfy the CBF conditions is a safe controller (see Ames’ CBF papers). Our paper learns the CBF $h$ and controller $u$ *at the same time.* However, when the environment is too complex and the learned controller hasn’t seen the current scenario, it may not perform well. In this case, the learned CBF is still valid and can be used as a guidance to update the control input u (computed by the policy network $\pi_i$) to ensure safety. To be concrete, at test time, if the CBF condition is not satisfied using current $h_i$ and $u_i$, we seek to compute a small $e_i$ such that $u_i+e_i$ can make sure that $h_i$ and $u_i+e_i$ satisfy the CBF conditions. The $e_i$ is computed by minimizing the $\phi(e_i)$ in Equation (9). We also added some detailed explanations in the brown part inSection 4.3, and added an ablation experiment studying the effects of the online policy refinement step.   Please also see our response to Reviewer 2 Question 6 for more discussions.
>
> **Q2: Why does the proposed method outperform baselines when the number of agents is large?**
>
> A2: Our control barrier functions and controllers are *fully decentralized* in both training and testing, and thus is not affected by the increasing number of agents as long as the overall density of the agents in the environment remains similar. The baseline methods MAMPS, MADDPG and PIC used *centralized* critic networks (which is common in MARL literature), which have to handle an exponentially growing state space when the number of agents increases. We think N=32 is simply a coincidence mainly because we chose the power of 2 as the number of agents in the experiments. In literature, very few papers reported *centralized* safe control for more than 50 nonholonomic systems.
>
> **Q3: Add the comparison to <Guaranteed Obstacle Avoidance for Multi-Robot Operations With Limited Actuation: A Control Barrier Function Approach>.**
>
> A3: We actually tried to run the code provided by the authors, but when the number of agents is greater than 8, it could take unaffordable time to do the computation. Furthermore, their method assumes that the CBF $h_i$ is perfectly given (handcrafted, not learned), while in our setup when the model dynamics is complex, constructing a handcrafted CBF is almost impossible. This was in fact one of the most important motivations of our present work. We have added a comparison with this approach on computational time to the Drones paragraph in Section 5.
>
> **Q4: Should the union bound in Equation (6) be a $\cup$ rather than $\cap$.**
>
> A4: It should be $\cap$. Equation (6) says that the probability that the generalization error $\epsilon_i$ is bounded for *all agents* is greater than $1-\delta$. So we take the intersection $\cap$.
>
> **Q5: Is the proposed method a supervised method rather than RL?**
>
> A5: Yes, your understanding is correct. Our method can be viewed as a supervised method. The supervision comes from the CBF conditions in (2). The updated policy will influence the data we collect because our method is not RL-based, and it is instead based on the dynamical model.
>
> **Q6: How to know the model dynamics $f_i$ in general?**
>
> A6: There are standard forms of $f_i$ for nonholonomic vehicles and drones used in our experiments. The $f_i$ that we have used are shown in **Appendix C** and they are all very standard models of the vehicles. This physical model is usually not hard to know for these practical control systems and is created using physical laws and classical mechanics such as Lagrangian Equations of Motion. If the $f_i$ is unknown, it is possible to fit the model from data first, and apply our approach, as in for example <Learning for safety-critical control with control barrier functions>. Getting $f_i$ is a research domain called *system identification*, which  is beyond the scope of our paper, and has been listed as one of our immediate future work.
>
> Thanks again for your supportive comments. Hope our explanation has successfully cleared your concern.

---

> ### Author Response · Authors · 2020-11-18
> **Thanks for raising the score to 8**
>
> We really appreciate the feedback from the reviewer, and thanks for raising the score!

---

### Official Review · AnonReviewer2 · 2020-10-30
**Sound theory, improvable experiments**

**Rating:** 8
**Confidence:** 2

**Review:**

## Post-rebuttal
The rebuttal was convincing (see my comment below) and addressed my concerns. I therefore increase my score to 8.

## Overview

This paper introduces MDBC, a method relying on a Control Barrier Function (CBF) learnt jointly with a policy, to obtain a safe and generalizable behavior.
It proceeds to show experiments on 2D and 3D navigation tasks, where the safety is defined as the avoidance of any collision.

The paper is globally well written, and the underlying theoretical foundations seem sound, and experiments show strong results.
However, the experiments leave out a lot of critical implementation details, raising concerns for the reproducibility of the work. Moreover, the baselines used seem inadequate.


## Method

The theory underpinning the proposed method is well exposed and sound.
The only question that I have is with respect to the choice of the neural network architecture. A PointNet is being used, what motivates this choice? Have you considered other alternatives such as Pointer network, Graph neural net, transformers?

## Experiments

Some information is missing to fully reproduce the experiments, including:
* The optimization parameters (optimizer, lr, and possible additions like weight decay and so on)
* The exact loss used for the policy training (the text mentions a "distance", but doesn't specify which
* How the policy loss and the safety loss $\mathcal{L}^c$ are combined (I would expect a weighted sum, but this is not specified as far as I can tell)
* Values of some other hyper-parameters, such as $\gamma$, $\mu$, $\lambda$. Even the non-linear activation $\sigma$ used in the network is unspecified
* The theory section assumes that each agents observe the states of other agents within a neighborhood. In the context of the experiments, this neighborhood is unspecified.


As for the comparison, I find the choice of some of the baselines inadequate. With the exception of MAFACTEST (which is, unexpectedly, not reported on all environments), all 3 others (PIC, MAMPS, and MADDPG) are reward based, while the proposed method uses policy imitation to a target policy (LQR or PID controller), which is a much simpler learning problem, as it sidestep many hard problems in RL such as exploration and credit assignment (both temporal and agent-wise). As such, it is not surprising that RL based approaches struggle more when trying to generalize/scale, especially if the training scenario always contain the same number of agents (better results could possibly obtained with some limited curriculum learning)

It is not even clear from the text how the rewards for these baseline are engineered in some environments like "Nested Rings" where the agents must additionally follow a predefined trajectory.

In addition, the safety loss used in the proposed approach to avoid collisions is enforced with some margin $\gamma$, which naturally encourages more conservative policies. By contrast, the reward provided to the RL baseline agents is negative only in the event of a collision (positive when the goal is reach, 0 otherwise). To make a fair comparison, the negative reward should be given whenever two agents are getting too close to each other, similar in spirit to the margin $\gamma$, so as to encourage more conservative policies in this case as well.

All in all, better baselines could probably devised, perhaps even as simple as well-tuned potential fields.

Aside from the baselines, the results shown are mostly convincing, and demonstrate good generalization capabilities. Some limitations:
* Even though results are averaged over 10 independent runs, now error bar are represented in any of the graphs, so we don't know to what extent the methods are brittle or not
* Section 4.3 introduces the "spontaneous online policy improvement" which is a test-time refinement of the policy to increase safety. However, no discussion of this component is made in the experimental section. In particular, what would be the performance without such refinement? The refinement requires doing some gradient descent over some cost function. How many step does that usually take? What is the impact on the run-time? What is the percentage of states/actions that need to be refined in practice (I expect this percentage to increase with the number of agents, perhaps the generalization capabilities can be credited solely to this component of the method? )
* It would be valuable to provide some sort of visualization of the $h$ function that has been learnt. How well does it correlate to the distance to the closest object?

---

> ### Author Response · Authors · 2020-11-16
> **Response to Reviewer 2**
>
> Thanks for finding our paper well-written and our theoretical results sound. The main issues regarding implementation details and baselines are addressed as follows. We have updated the paper by adding the implementation details. We have also provided the anonymous code to the reviewer for reproducibility check.
>
> **Q1 (Method): What motivates the use of PointNet?**
>
> A1: There are mainly two reasons. 1) PointNet is *invariant* to the size and permutation of the input data, thus can be used to encode the observation (states of surrounding agents) that has *time-varying* quantity and permutation. 2) The structure of PointNet is very simple comparing to other networks, and is sufficient to meet our need of encoding quantity-permutation invariant property. This quantity-permutation invariant property is essential to our prominent *scalability*. We hope to keep the network simple because our main contribution is not to design a neural network. Instead, we focus on learning safe multi-agent control via decentralized CBF.
>
> **Q2: In the experiment, some information is missing.**
>
> A2: We have updated our submission (see the brown part of Section 4.1, 4.2, 4.3, and 5) and added the missing information. The learning rate, weight decay, loss for policy training, and the combination of all losses are specified in Section 4.1. The activation function is specified in Section 4.2. The values of other hyper-parameters are given in Section 4.1 and 4.3 correspondingly. The definition of observation radius is added to the third paragraph (Evaluation Criteria) in Section 5. Thank you for pointing out the lack of information and providing the detailed instructions.
>
> **Q3: The comparison to some baselines is inadequate.**
>
> A3: We believe our comparison to the model-free methods is fair: Although our method imitates the LQR controllers for goal-reaching, in training the RL-based methods, the control action computed by LQR is also fed to the agent as one of the inputs to the actor-network. So the RL agents can also learn to use LQR as a reference for goal-reaching. Since the model-free methods do not have access to model dynamics and are more data-demanding, the number of state-observation pairs to train MADDPG and PIC is 10^3 times more than that of model-based learning methods to make sure they converge to their best performance (see the second paragraph of Section 5).
>
> Moreover, we believe that imitating the LQR controller does *not* make the problem much simpler because our main focus is safety rather than goal-reaching, and the goal-reaching controller never takes into account collision-avoidance. The goal-reaching controller is to make sure that our learned controllers make the agents collision-free *without being deviated from its original tasks too much* (i.e. goal-reaching in this case). This goal-reaching controller can be replaced by other controllers to accomplish other tasks.  We totally agree that with some curriculum learning added, the RL based methods may perform better. However, our message here is simply to show that in terms of “safety guarantees”, these “reward-based” methods are not that favorable (though many MARL works have used this collision-based/distance-based reward engineering in the literature).
>
>
> **Q4: How is the reward of Nested Rings engineered?**
>
> A4: In addition to the reward terms specified in the first 3 paragraphs of Section 5, we also give the agent a negative reward proportional to the distance to the nearest point on the rings. This would encourage the agents to stay close to the ring-like trajectories.
>
> **Q5: The $\gamma$ factor can make the comparison unfair because our method is more conservative.**
>
> A5: When training other baseline methods, we have also bloated the dangerous set to make the agents less aggressive, and fine-tuned the bloating factor to achieve a reasonable trade-off between the safety rate and reward. Therefore, we believe we are not introducing any unfair comparison by using $\gamma$.
>
> **Q6: The error bars should be added. The online policy refinement should be discussed in the experiment.**
>
> A6: Thanks for finding our results “convincing” and “demonstrating good generalization capabilities”. We have added the error bars in Figure 4 and Figure 5. An ablation study on online policy refinement is also added to **Appendix D.** The performance is still promising without online policy refinement. The refinement further pushes the boundaries of the safety rate. In general, less than 10 iterations of refinement is needed to satisfy the CBF conditions, which has a negligible influence on the run time. The percentage of steps requiring the refinement will not exceed 17% even when there are 1024 agents. We have also added the motivation of the online policy refinement in Section 4.3.
>
> **Q7: Visualization of the CBF (h function).**
>
> A7: Yes, we have added the visualization of the learned CBF in Figure 9 of Appendix D and the analysis in the last paragraph of **Appendix D.**

---

> > ### Comment · AnonReviewer2 · 2020-11-18
> > **Thank you for the rebuttal**
> >
> > I would like to thank the authors for their very thorough rebuttal. All my concerns have been satisfactorily addressed, and I believe the paper's strength has been significantly improved, especially from a reproducibility standpoint.
> >
> > In particular two key details that were originally left out (namely "in training the RL-based methods, the control action computed by LQR is also fed to the agent as one of the inputs to the actor-network" and "we also give the agent a negative reward proportional to the distance to the nearest point on the rings") are now added to the text and I agree with the authors that it makes the comparison with reward-based method fair and sound.
> >
> > I also enjoyed the added visualizations and ablations in Appendix D, and found figure 9 particularly insightful.
> >
> > Finally, the added error bars show that the method is quite robust, whereas the baselines show signs of brittleness. This is a good sign for the applicability of the proposed method.
> >
> > For all these reasons I will raise my score to 8.

---

> ### Author Response · Authors · 2020-11-19
> **Thanks for raising the score to 8!**
>
> We would like to thank the reviewer for raising the score to 8! We also appreciate the valuable comments, which helped us significantly improve the paper's strengths.

---

### Official Review · AnonReviewer5 · 2020-11-06
**Interesting approach; experiment details missing**

**Rating:** 7
**Confidence:** 3

**Review:**

This paper brings the recently introduced idea of control barrier functions (CBF) as safety certificate to the multi-agent land. To this end it introduces the idea of decentralized CBFs. This is followed with a framework for jointy learning the CBFs and policies with a PointNet inspired network architecture. The paper provides some generalization guarantees for the approach as well. Finally the approach is compared against various learning and planning based baselines where it significantly outperforms.

Overall, it's a very interesting extension of the CBF idea for the decentralized setting and permutation+#agents invariant setup seems quite scalable. It's good to have proven generalization guarantees as well.

Although the paper shows impressive performance on a variety of tasks, the learning setup is unclear. Sure you have the loss function in Sec 4.1 with the computational graph in Fig 1. But it's unclear how exacly learning happens? Like how is the data collected; how are the enviornments explored. Do you have to estimate policy gradients or you have exact gradients. How exactly do you compare against both MADDPG style model-free method as well as planning based MAFACTEST. Do they even have the exact same environment models? The details are unclear, so it's not possible for me to really judge what's going on in Fig 4. Sure the average rewards and safety rates are higher but are the quantities even commensurable for these very different methods? It's probably yes gven the quantified metrics are scalar quantities which can possibly be measured in a similar manner for the approaches. But still more information is necessary for it be meaningful.

---

> ### Author Response · Authors · 2020-11-16
> **Response to Reviewer 5**
>
> Thanks for finding our work very interesting, acknowledging our theoretical results, and finding our empirical results impressive. In the following, we will address the two main issues that you are concerned about. We appreciate that you pointed out the lack of setup information, and we have updated our draft accordingly. We hope that you find the updated information improving the paper and making the setup clear.
>
> **Q1: The learning setup of our method is unclear.**
>
> A1: We have updated Section 4.1 (see the highlighted part) to give a detailed description of how the data is collected, how the environment is explored, how the gradients are computed, and how the network is trained. In training, all agents are put into the specific environment, which is not necessarily the same as the testing environment, to collect state-observation pairs $(s_i, o_i)$ under their current policies with probability $1-\iota$ and random policies with probability $\iota$, where $\iota$ is set to be 0.05 in our experiment. The collected $(s_i, o_i)$ are stored as a temporary dataset and in every step of policy update, 128 $(s_i, o_i)$ pairs are randomly sampled from the temporary dataset to calculate the total loss $\mathcal{L}$. We minimize $\mathcal{L}$ by applying stochastic gradient descent to the parameters of CBF and control policy.
>
> **Q2: The configuration of baseline approaches is unclear.**
>
> A2: We have updated the first two paragraphs of Section 5 to explain the configurations of model-based MAFACTEST and model-free MADDPG and PIC. The evaluation criteria, including how the safety rate and reward are computed, is presented in the third paragraph of Section 5, i.e., the **Evaluation Criteria** paragraph.. Please see the highlighted parts. In short, for each task, the environment model is the same for all the methods. The exact model dynamics are visible to model-based methods including MAMPS, MAFACTEST and our methods, and invisible to the model-free MADDPG and PIC. Since the model-free methods do not have access to model dynamics but instead the simulators, they are more data-demanding. The number of state-observation pairs to train MADDPG and PIC is $10^3$ times more than that of model-based learning methods to make sure that they converge to their best performance.
>
>
> Thanks again for the valuable comments. We hope our additional explanation of the experimental setup has cleared the concern. Please also see our response to Reviewer 2 Question 2 and 3 for more explanations. As the comments are only concerning presentation (instead of technical parts or contribution), we sincerely hope that the reviewer can re-evaluate our paper after seeing the updated version. More comments on further improving the presentation are also very much welcomed.

---

> > ### Comment · AnonReviewer5 · 2020-11-19
> > **Thanks for the update**
> >
> > I'm satisfied with the updated descriptions. Still it would be great if authors plan to release their source code to the community.

---

> > > ### Author Response · Authors · 2020-11-20
> > > **Thanks for raising the score to 7!**
> > >
> > > We would like to thank the reviewer for raising the score to 7! We plan to open source the code to the community.

---

### Author Response · Authors · 2020-11-16
**General remark**

We first thank all the reviewers for their constructive and valuable comments. We have noticed that many of the reviews are quite positive, finding our work “very interesting”, “theoretical results solid”, “empirical results impressive”, and “well-written” in general. We really appreciate those positive comments. It seemed a pity that most of the negative comments came from our missing information/insufficient explanation on the experimental settings, and elaboration of our contribution. We have updated the submission as per the reviews, by adding more detailed explanations on the settings and our contribution. We have also updated some experimental results as suggested by the reviewers. We sincerely hope that the reviewers can re-evaluate our paper after reading our updated version. More discussions and suggestions on further improving the paper are also always welcomed!

---

> ### Author Response · Authors · 2020-11-18
> **Status update**
>
> We thank Reviewer 4 for the timely response, and raising the score to 8. We are looking forward to the feedback from other reviewers after seeing our rebuttal and revision, and more discussions. Thanks!

---

> ### Author Response · Authors · 2020-11-19
> **Status update**
>
> We would like to thank Reviewer 2 for raising the score to 8! Now we have two 8s.
> We sincerely hope that our response and detailed revision can also address other reviewers’ concerns. We really appreciate it if they could also re-evaluate the submission. Thanks!
>
> We appreciate Reviewer 5's comments and raising the score to 7!
>
> Thanks to Reviewer 1 for raising the score to 6! We look forward to the reply from Reviewer 3.

---

### Author Response · Authors · 2020-11-25
**General remark after rebuttal**

We appreciate the reviewers for their valuable comments. We also really appreciate that 4 out of 5 reviewers have raised their score to “strong accept” (8), “accept” (7) and “weak accept” (6) during the rebuttal phase, after we added the important missing information on the experimental setup and supplementary experiments. We understand that reviewer 3 may not have time to check our detailed (follow-up) response and revision before the rebuttal deadline. However, we sincerely hope that reviewer 3 and the area-chair can re-evaluate our work based on the updated information. We believe we have made the theoretical part sound and self-contained. More importantly, we are confident that this is a breakthrough in learning safe multi-agent system control. To the best of our knowledge, there is no existing work that can safely control >1000 nonholonomic systems in 3D while providing real-time execution capability. Most existing methods either can only handle <20 quadcopters or with safety ratios dramatically deteriorating as the number of agents increases. Our method has demonstrated significant improvements in terms of both the safety rate and the total reward, and our novelty and contribution have been well-acknowledged by the positive reviews.

We are grateful for the reviewers’ suggestions that have helped us greatly improve the quality of our submission. We hope that the area-chair would consider the strong recommendations by the positive reviewers which are concluded based on our detailed discussions.

---

### Decision · Program_Chairs · 2021-01-07
**Final Decision**

**Decision:**

Accept (Poster)

**Comment:**

Initially there were some shared concerns about the work being too incremental, lack of technical clarity on the algorithmic side and experiments, and lack of clear mathematical formulations. The authors did a good effort and cleared up many questions and remarks satisfactorily, and several reviewers have increased their scores as a consequence. In its current state I recommend to accept the paper.